# Red Dragon Fruit Peels: Effect of Two Species Ratio and Particle Size on Fibre Quality and Its Application in Reduced-Fat Alpaca-Based Sausages

**DOI:** 10.3390/foods13030386

**Published:** 2024-01-24

**Authors:** Andrés A. Corimayhua-Silva, Carlos Elías-Peñafiel, Tatiana Rojas-Ayerve, Américo Guevara-Pérez, Lucero Farfán-Rodríguez, Christian R. Encina-Zelada

**Affiliations:** 1Departamento de Tecnología de Alimentos, Facultad de Industrias Alimentarias, Universidad Nacional Agraria La Molina (UNALM), Av. La Molina s/n Lima 12, Lima 15024, Peru; 20181617@lamolina.edu.pe (A.A.C.-S.);; 2Departamento de Química, Facultad de Ciencias, Universidad Nacional Agraria La Molina (UNALM), Av. La Molina s/n Lima 12, Lima 15024, Peru; 3Instituto de Investigación de Bioquímica y Biología Molecular (IIBBM), Universidad Nacional Agraria La Molina (UNALM), Av. La Molina s/n Lima 12, Lima 15024, Peru

**Keywords:** *Hylocereus hybridum*, *Hylocereus undatus*, pitahaya, *Vicugna pacos*, meat products, FT-IR spectra, texture profile analysis, nine-point hedonic scale, 3^k^ full factorial design, response surface methodology

## Abstract

This research aimed to assess the influence of red dragon fruit peels ratio (RDF-PR) from two species, *Hylocereus hybridum* (HH) and *Hylocereus undatus* (HU), and particle size (PS) on quality parameters of red dragon fruit peel powder (RDF-PP) and its further application in emulsified alpaca-based sausages as partial substitutes of pork-back fat. A three-level full factorial design (nine treatments) was employed to evaluate the effect of RDF-PR (HH(0%):HU(100%), HH(50%):HU(50%), and HH(100%):HU(0%)) and PS (499–297, 296–177, and <177 µm) on the dependent variables: L*, a*, b*, C, h°, water-holding capacity, oil-holding capacity, swelling capacity, pectin yield, degree of esterification (analysed through FT-IR), and crude fibre content. The data analysed through a response surface methodology showed that treatment one (T1) is the best with the optimised conditions at 100% HU RDF-PR and PS of <177 µm. The statistical validation of T1 exhibited the highest water-holding capacity (32.1 g/g peel), oil-holding capacity (2.20 g oil/g peel), and pectin yield (27.1%). A completely randomised design (four formulations) was then used to assess the effect of partial replacement of pork-back fat by T1 in emulsified alpaca-based sausages on the colourimetric, physicochemical, and texture properties (hardness, chewiness, cohesiveness, springiness, adhesiveness, and adhesive force). Likewise, a sensory hedonic scale was employed to evaluate the appearance, colour, odour, flavour, texture, and overall acceptability of sausages. The results revealed that 65.7% of pork-back fat content was successfully replaced compared with a control formulation. Additionally, F3 showed significantly (*p* < 0.05) better colourimetric, physicochemical, and textural characteristics, such as lower hardness (34.8 N) and chewiness (21.7 N) and higher redness (a* = 19.3) and C (22.9), compared to a control formulation. This research presents RDF-PP as a promising fat substitute for developing healthier, reduced-fat meat products using fibre-rich agroindustry by-products.

## 1. Introduction

Emulsified meat products, such as sausages, are typically manufactured from various meat sources, including pork, beef, and chicken [1,2,3]. However, it is feasible to use alpaca meat, a South American camelid species reared for their fleece and meat in the Altiplano of Peru at 4000 m.a.s.l. [4,5]. Alpaca meat is a raw material with remarkably low intramuscular fat (<1.00%) and cholesterol (51.1 mg/100 g) contents, high protein content (22.7–23.3%), a high ratio of omega 6/omega 3 fatty acids (3.74), and a low lipid oxidation rate compared to other meats, such as beef and lamb [4,6,7]. Therefore, the consumption of alpaca meat has the potential to support small-scale producers and provide economic relief to high Andean families [4].

Emulsified sausages [8] typically contain 20–35% fat, primarily sourced from pork-back fat [2,8,9,10]. This animal fat is recognised for its high content of saturated fatty acids (33.9%) [11], which is linked to an increased risk of obesity, coronary heart disease, and certain types of cancer [12]. The World Health Organization stated that the consumption of foods rich in saturated fats has been steadily increasing [13], which is particularly concerning in industrialised countries, where obesity is estimated to affect ~13.2% of the world’s population by 2030 [2]. Given this context, the high fat and saturated fatty acids content of emulsified sausages holds a potential health risk and may discourage health-conscious consumers from choosing these products.

Several studies have proposed the utilisation of fibres to reduce fat content and improve the quality characteristics of various meat products without adverse effects on processing yield [2,14,15,16,17]. Fibre can be classified into insoluble (cellulose and hemicellulose) and soluble (pectin, galactomannan, inulin, and gum) fibres based on their water solubility [17]. Regarding soluble fibres, pectin is a complex structured polysaccharide that has received much attention as a food additive because of its health benefits, thickening, gelling, and emulsification properties [18,19]. In the food industry, pectin has the potential to serve as a fat substitute, providing a low-calorie and reduced-fat alternative [19]. Wongkaew et al. [17] demonstrated that mango peel pectin at a concentration of 5% successfully substituted animal fat in the development of dried Chinese sausages without impairing their texture. Likewise, Vilcapoma et al. [20] showed that pectin isolated from yellow dragon fruit peels replaced 78.0% of animal fat in low-fat emulsified alpaca-based sausages without altering their texture and colour. The incorporation of fibres in meat products not only enhances their nutritional profile but also improves quality attributes, including texture and cooking yield, through increased water- and fat-binding capacities [21,22]. Nonetheless, the techno-functional properties of soluble and insoluble fibres can vary depending on processing factors, such as grinding and drying [21].

Dragon fruit of the genus *Hylocereus* is a tropical fruit native to Central and South America. Its peel has a high content of pectin (from 9.62 to 30.2%), low lipid content (from 2.43 to 4.13%), and betacyanins (~151 mg/100 g) [23,24,25]. Unfortunately, the pre-treatment of dragon fruit in supermarkets and fruit processing plants generates a large amount of dragon fruit peel waste, which, consequently, represents an uneconomical activity and intensifies environmental problems by contributing to the development of microorganisms due to its high organic and nutritional value [18,26].

Recent studies have reported the use of dragon fruit peels in various food products, such as ice cream, bread, noodles, cookies, and sausages [20,23,25,27]. For instance, Aukkanit et al. [23] reported a 4.47% reduction in fat content by substituting pork-back fat with 0.50, 1.00, 1.50, 2.00, and 2.50% dragon fruit (*Hylocereus undatus*) peel powder in emulsion sausages. Accordingly, the authors developed a well-scored product with 2.00% RDF-PP with comparable sensory attributes to a control formulation. Additionally, Utpott et al. [28] achieved a 73.5% fat reduction in strawberry ice cream with high overall acceptability using 1.00% pitaya (*Hylocereus polyrhizus*) peel powder. On the other hand, studies also found the effect of particle size (PS) on the physicochemical and techno-functional properties of foods [25,29,30]. For instance, Mai et al. [25] reported a significant increase in soluble fibre content (1.66–1.94%) in cookies when using 10.0% pitaya (*Hylocereus undatus*) peel powder at 595, 420, 210, and 105 µm. However, to the best of our knowledge, studies on the application of dragon fruit peels with different PS and various ratios of peels from distinct species have not been reported. Likewise, there are no reports on the application of red dragon fruit peels as fat substitutes in emulsified alpaca-based sausages. In dragon fruit peels, Zhuang et al. [29] reported a significant increase in water-retention capacity, oil-retention capacity, and swelling capacity at a PS between 58 and 104 µm, compared to other fractions of different particle sizes: 104–178 µm and <58 µm.

Therefore, the use of dragon fruit peels from agroindustry residues as fat replacers could contribute to the development of healthier meat products with better-quality properties. The present research aimed to assess the effect of red dragon fruit (*Hylocereus hybridum* (HH) and *Hylocereus undatus* (HU)) peels ratio and particle size on the colourimetric, techno-functional, and physicochemical characteristics of red dragon fruit peel powder to evaluate the effect of partial replacement of pork-back fat by red dragon fruit peel powder on the colourimetric, physicochemical, and textural characteristics of emulsified alpaca-based sausages, and to perform a sensory evaluation through a hedonic scale to compare the developed reduced-fat sausage against a control formulation.

## 2. Materials and Methods

### 2.1. Materials

Red dragon fruits from *Hylocereus hybridum* (HH, ~55 kg; peel thickness: 0.45 cm; peel yield: 22.1%; MC: 91.1%; SS: 0.61 °Brix; TA: 0.13%) and *Hylocereus undatus* (HU, ~55 kg; peel thickness: 0.35 cm; peel yield: 23.9%; MC: 88.2%; SS: 0.63 °Brix; TA: 0.08%) species (Appendix A) were provided by the Corporation Abregú (latitude: −11.41694; longitude: −77.23363), Huaral, Peru. The fruits were harvested at the third flowering two days before transportation, packed in perforated carboard boxes, transported to the laboratory from the Universidad Nacional Agraria La Molina (UNALM), and stored at 4 °C until further analysis. Alpaca meat was procured from the Sociedad Agrícola de Interés Social, SAIS-Túpac Amaru (latitude: −11.782611; longitude: −75.716833), Junín, Peru. Pork-back fat was obtained from the UNALM Sales Center, Lima, Peru. Cure salt, sodium erythorbate, sodium tripolyphosphate, soy protein isolate, carmine colourant, and liquid smoke were obtained from Alitecno S.A.C., Lima, Peru, and *k*-carrageenan from Polifood Peru S.A.C., Lima, Peru. Salt, garlic powder, onion powder, and spices were purchased from the local market.

### 2.2. Production of Red Dragon Fruit Peel Powder

Red dragon fruit peel powder (RDF-PP) was obtained according to the method of Aukkanit et al. [23]. Peels of HH and HU (Appendix A) were separated from the pulp, areole, and flower end pit. Peels were then sliced in a cutting machine (CL50 Ultra, Robot Coupe, Dijon, France) to a 0.15 ± 0.05 cm thickness and dried in a hot air tray dryer (DRR-200, Reter) at 60 ± 2 °C for 18 h or until constant weight was achieved. Finally, dried peels with a final moisture content of ~5.20% were milled in a rotor beater mill (SR 300, Retsch GmbH, Haan, Germany) at 3500 rpm, equipped with a 0.5 mm mesh screen. The subsequent RDF-PP fractions were obtained using a RO-TAP test sieve shaker (RX-29-16, WS Tyler, Mentor, OH, USA) equipped with standardised test sieves N° 35 (500 µm), N° 50 (297 µm), and N° 80 (177 µm) based on a 3^k^ full factorial design (Figure 1).

### 2.3. Determination of the Physicochemical and Techno-Functional Properties of RDF-PP

#### 2.3.1. Colour Measurement

Colour tristimulus parameters based on the CIEL*a*b* system and additional colour characteristics, such as chroma (C) and hue angle (h°), were measured using a colourimeter (NR200, 3nh, Guangzhou, China). The L* value is a measure of lightness that ranges from black (0) to white (100). The coordinate a* ranges from red (+) to green (−), and b* ranges from yellow (+) to blue (−). C [√ (a*^2^ + b*^2^)] indicates the purity or saturation of colour, and h° (tan^−1^ b*/a*) indicates the sample colour (0° = red-purple, 90° = yellow, 180° = bluish green, 270° = blue) [31]. The instrument settings were light source D65 and measuring aperture (Ø) 8 mm.

#### 2.3.2. Crude Fibre Content

The crude fibre (CF) content of RDF-PP was determined according to the AOAC 978.10 official method [32]. In an Erlenmeyer flask, 200 mL of 1.25% (*w*/*v*) H_2_SO_4_ solution, preheated to 98 ± 1 °C, was added to 2 g of sample. The suspension was boiled for 30 min with continuous stirring at 417 rpm, using manual shaking at 5 min intervals to prevent solid adherence to the glass. The suspension was then vacuum filtered and washed with 200 mL of distilled water and preheated to 99 ± 1 °C. The filtered residue was collected to the Erlenmeyer flask containing 200 mL of 1.25% (*w*/*v*) NaOH solution, preheated to 98 ± 1 °C, and boiled again. After digestion, the vacuum-filtered residue was washed with 30 mL of 1.25% H_2_SO_4_ solution at 98 ± 1 °C and 100 mL of distilled water at 99 ± 1 °C. Subsequently, the residue was dried in an oven (UN110, Memmert, Schwabach, Germany) at 130 °C for 2 h, cooled in a desiccator, and weighed (W_1_). Once dried, the residue was incinerated in a muffle furnace (ECO 110/3, Protherm, Batikent, Turkey) at 550 ± 10 °C for 2 h, followed by cooling in a desiccator. The weight of the incinerated residue was recorded (W_2_), and the CF was calculated using the following equation:(1)$$CF\left(\%\right)=\frac{W_{1}-W_{2}}{W_{i}}\times 100$$
where ‘W_1_′ represents the weight (g) of the dried sample residue, ‘W_2_′ represents the weight (g) of the incinerated sample residue, and ‘W_i_’ represents the initial weight (g) of the sample.

#### 2.3.3. Pectin Yield

The pectin yield (PY) of RDF-PP was determined using the method described by Zahari et al. [24]. In an ISO glass bottle, RDF-PP was extracted with water in a 1:50 (*w*/*v*) ratio at pH 2.0 ± 0.03 using citric acid with continuous stirring at 150 rpm in a water bath (MaXturdy 18, Daihan Scientific, Gangwon-do, Korea) at 68 ± 0.4 °C for 77 min. After cooling, the suspension was filtered (Whatman N° 113) and concentrated to 50% of its initial volume in a rotary evaporator (RE301, Yamato Scientific, Tokyo, Japan) at 55 °C, 81 rpm, and 90 kPa. For the precipitation of pectin, 96% ethanol (*v*/*v*) was added to the concentrated solution at a 1:2 (*v*/*v*) ratio, and the mixture was kept at 4 °C overnight. Next, the uppermost pectin layer was subjected to filtration and underwent a series of three washes using ethanol solutions at 50, 75, and 100%. Then, the retained pectin was dried in an oven at 50 °C until constant weight, cooled, ground, and weighed. The PY was calculated using the following equation:(2)$$PY\left(\%\right)=\frac{P}{W_{i}}\times 100$$
where ‘P’ represents the weight of dried pectin (g) and ‘W_i_’ represents the initial weight of the sample (g).

#### 2.3.4. Degree of Esterification

The degree of esterification (DE) was determined according to Muhammad et al. [33] by analysing the chemical structure of pectin obtained from dragon fruit peels using an FT-IR spectrometer (Spectrum Two, PerkinElmer, Waltham, MA, USA). The pectin sample was constrained on the UATR diamond crystal using a pressure tower. Subsequently, the FT-IR spectra were determined at the transmittance mode in the frequency range of 900–4000 cm^−1^ (mid-infrared region) at a resolution of 4 cm^−1^ with 128 scans. The absorption bands at 1630 cm^−1^ (A_1630_) and 1745 cm^−1^ (A_1745_) were used to calculate the DE using the following equation:(3)$$DE\left(\%\right)=\frac{A_{1745}}{A_{1745}+A_{1630}}\times 100$$
where ‘A_1745_′ and ‘A_1630′_ represent the absorption bands at 1630 and 1745 cm^−1^ for non-methyl-esterified carboxyl and methyl-esterified carboxyl groups, respectively.

#### 2.3.5. Water- and Oil-Holding Capacity

Water-holding capacity (WHC) and oil-holding capacity (OHC) were determined according to Chia and Chong [31]. In a centrifuge tube, 15 mL of distilled water or commercial soybean oil was added to 0.15 g of RDF-PP, followed by stirring for 30 s at 800 rpm. After that, the suspension was allowed to stand for 1 h at room temperature. The suspension was then centrifuged at 5000 rpm and 14 °C for 6 min. Once the supernatant was carefully discarded, the weight of the sediment was recorded and used to calculate the WHC or OHC using the following equation:(4)$$WHC\;or\;OHC=\frac{W_{s}-W_{i}}{W_{i}}$$
where ‘W_s_’ represents the weight (g) of the sediment (sample + water or oil) and ‘W_i_’ represents the initial weight (g) of the sample.

#### 2.3.6. Swelling Capacity

The swelling capacity (SC) of RDF-PP was estimated according to Chia and Chong [31]. In a 50 mL test tube, 1 g of sample was added to 30 mL of distilled water while stirring at 600 rpm to avoid a lumpy suspension. Next, the volume was then increased to 50 mL, and the suspension was stirred at 1000 rpm for 20 s. The suspension was allowed to stand for 20 h at room temperature to thoroughly hydrate the sample. The initial volume (V_1_) and final volume (V_2_) occupied by the sample were recorded to calculate SC using the following equation:(5)$$SC\left(\frac{mL}{g}\right)=\frac{V_{2}-V_{1}}{W_{i}}$$
where ‘V_1_′ represents the initial volume (mL) occupied by the sample in the tube, ‘V_2_′ represents the final volume (mL) occupied by the swollen sample in the tube, and ‘W_i_’ represents the weight (g) of the sample.

#### 2.3.7. Particle Size Distribution

The PS distribution of RDF-PP of the best treatment obtained from the 3^k^ full factorial design (Figure 1) was measured using the image analysis method described in ASTM F1877-16 [34]. In brief, multiple images of RDF-PP dispersed in a dry environment were taken using a light microscope (Olympus X85, Olympus, Tokyo, Japan), combined with a digital camera (Nikon, Tokyo, Japan), and analysed using ImageJ software (version 2.1). The PS distribution was characterised by the average PS (X_PS_), equivalent diameter at a cumulative frequency of 50% (D_50_), and equivalent diameter at a cumulative frequency of 90% (D_90_).

### 2.4. Production of Emulsified Alpaca-Based Sausages

The emulsified alpaca-based sausages were prepared according to the method described by El-Nashi et al. [1]. Boneless and connective tissue-free alpaca (*Vicugna pacos*) meat (*Biceps femoris* muscle from a 24-month-old male alpaca) and pork-back fat were cut into pieces (3 × 3 cm), frozen at −18 °C, and ground separately. Then, in a food processor (Moulinex, Paris, France), the pork-back fat (15.0%) was pre-emulsified with cold water (15.0%) and soy protein isolate (3.00%). The emulsified mixture was achieved by (1) blending the alpaca meat (58.7%), sodium chloride (2.26%), curing salt (0.05%), sodium erythorbate (0.09%), sodium tripolyphosphate (0.35%), and ice (0.66%) for 5 min at a temperature below 4 °C; (2) pre-emulsified pork-back fat (30.0%), ice (1.34%), *k*-carrageenan (0.01%), garlic powder (1.00%), onion powder (1.20%), liquid smoke (0.10%), carmine colourant (0.05%), and spices mixture (1.15%) were added, taking care that the temperature was below 15 °C. Subsequently, pork-back fat substitution with hydrated RDF-PP was performed at 3.29 (F1), 6.57 (F2), and 9.86% (F3) based on the whole formulation (i.e., 100%). Specifically, in order to develop the different fibre-enriched alpaca-based sausage formulations (F1, F2, and F3), in the pre-emulsified mixture, the pork-back fat (15.0%) was replaced by 21.2 (F1), 42.5 (F2), and 63.7% (F3) RDF-PP and 0.67 (F1), 1.33 (F2), and 2.00% (F3) water, respectively. The incorporation of RDF-PP and water into the formulations F1, F2, and F3 was based on the WHC of RDF-PP from the best treatment. Following this, the emulsified mixture was stuffed into 22 mm diameter collagen casings, portioned into approximately 15 cm lengths, and then subjected to heat treatment at 77 °C until the core temperature reached approximately 73 °C. Finally, the sausages were cooled immediately in cold water at 4 °C and vacuum-packed in polyethylene bags for storage at 4 °C.

### 2.5. Physicochemical Properties of Emulsified Alpaca-Based Sausages

#### 2.5.1. Colour Measurement

Alpaca-based sausage samples were sliced to a thickness of 1 cm. Then, colour parameters (L*, a*, b*, C, and h°), based on the CIEL*a*b* system, were measured ten times on the surface of each sample [17].

#### 2.5.2. Physicochemical Properties

Moisture (950.46), fat (991.36), protein (984.13), and total dietary fibre (991.20) contents were analysed according to the AOAC official methods [32]. Water activity was determined using an Aqualab 4TEV water activity metre (Decagon Devices, Pullman, WA, USA) [35].

#### 2.5.3. Frying Loss and Cooking Yield

Frying loss (FL) was determined according to the method of Amini-Sarteshnizi et al. [36]. Sausages sliced to a thickness of 1 cm were deep-fat fried at 178 ± 2 °C for 2 min until a core temperature of 73 ± 1 °C was obtained. The sausages were then allowed to cool at room temperature. The FL was calculated using the following equation:(6)$$FL\left(\%\right)=\frac{W_{sb}-W_{sa}}{W_{sb}}\times 100$$
where ‘W_sb_’ and ‘W_sa_’ represent the weight of the sausage before and after cooking, respectively. Cooking yield (CY) was determined by subtracting the FL value from 100%.

#### 2.5.4. Texture Profile Analysis

The texture profile analysis (TPA) of the sausages was conducted according to the method of Grigelmo-Miguel [37]. Slices 1.8 cm in diameter and 1.5 cm in thickness were cut from three different sausages. A texture analyser (CTX, Brookfield Ametek, MA, USA) was used to compress each slice with a trigger force of 0.07 N at a speed of 2 mm/s using a 5 kg load cell. Then, compression was conducted in two consecutive cycles at 50% of each slice’s original height, with an 11 s interval between cycles. A time–force graph was plotted, and hardness, chewiness, cohesiveness, springiness, adhesiveness, and adhesive force were recorded.

### 2.6. Sensory Evaluation of Emulsified Alpaca-Based Sausages

Emulsified alpaca-based sausage samples were removed from the casings, sliced into 1 cm thick cylinders, and randomly assigned a three-digit code for each formulation (C, F1, F2, and F3). Two sausage slices, constituting a serving size of ~8 g per formulation, were arranged on a disposable plate and served at room temperature to each panellist. Additionally, each panellist was given a testing time of 9 min and instructed to rinse their mouth with water after evaluating each sample. The optimal sausage sample against a C of emulsified alpaca-based sausages with 0.00% hydrated RDF-PP was assessed by 75 untrained panellists (16–43 years old). The panellists were instructed to evaluate the following attributes: appearance, colour, odour, flavour, texture, and overall acceptability. Each attribute was rated using a 9-point hedonic scale, ranging from 1 (i.e., ‘extremely dislike’) to 9 (i.e., ‘extremely like’) [38].

### 2.7. Statistical Analysis

#### 2.7.1. Three-Level Full Factorial Design and Response Surface Methodology

For the analyses of physicochemical and techno-functional properties of RDF-PP, nine treatments (Appendix A) were obtained through a 3^k^ full factorial design (Figure 1) using the softwares of STATISTICA (v. 10.0, Tulsa, OK, USA) and Design Expert (v. 9.0.6.2, Stat-Ease Inc., Minneapolis, MN, USA), both at a significance level of 5% (α = 0.05). Three levels were determined for each of the two factors: (i) red dragon fruit peels ratio (RDF-PR; HH(0%):HU(100%), HH(50%):HU(50%), and HH(100%):HU(0%)) and (ii) particle size (PS; coarse fraction: 499–297 µm, intermediate fraction: 296–177 µm, and fine fraction: <177 µm).

The response surface methodology was employed to generate the response surface plots for each dependent variable (L*, a*, b*, C, h°, WHC, OHC, SC, PY, CF, and DE), including R^2^, adjusted R^2^, and lack of fit, following the polynomial model:(7)$$y=\beta_{0}+\sum_{i=1}^{k}\beta_{i}x_{i}+\sum_{i=1}^{k}\beta_{ii}x_{i}^{2}+\sum_{i<j}\beta_{ij}x_{i}x_{j}+\varepsilon$$
where ‘y’ represents the dependent variable (L*, a*, b*, C, h°, WHC, OHC, SC, PY, CF, or DE); ‘β_0_′ represents the intercept term; ‘β_i_’ represents the linear coefficients for the main effects; ‘β_ii_’ represents the quadratic coefficients for the squared terms; ‘β_ij_’ represents the interaction coefficients for the cross-product terms; ‘x_i_’ represents the first independent variable (RDF-PR); ‘x_j_’ represents the second independent variable (PS); ‘k’ represents the number of dependent variables (RDF-PR and PS); and ‘ε’ represents the error term.

Before statistical validation, three (*n =* 3) dependent variables (WHC, OHC, and PY) of RDF-PP were optimised to obtain a geometric mean of the individual desirability functions (D). As indicated in Equation (8), each dependent variable (d_1_, d_2_, etc.) is converted into an individual desirability function (d_n_) that varies from 0.0 to 1.0. Thus, by setting d_n_ = 1, the adequate factor (RDF-PR and PS) levels can be achieved by maximising the response variable value [39].
(8)$$D=\sqrt[{n}]{d_{1}\times d_{2}\times \ldots \times d_{n}}$$
where ‘D’ represents the geometric mean of the individual desirability functions, ‘d_1_′ represents dependent variable 1, ‘d_2_′ represents dependent variable 2, ‘d_n_’ represents dependent variable ‘n’, and ‘n’ represents the number of response variables to optimise.

A validation analysis at a 95% confidence level (1 − α = 0.95) was performed to compare the experimental vs. theoretical values of WHC, OHC, and PY. A *p*-value exceeding 0.05 (*p* > 0.05) denotes the absence of statistically significant differences between the experimental and theoretical values.

#### 2.7.2. Completely Randomised Design (CRD)

For the analyses of the physicochemical properties of emulsified alpaca-based sausages, a completely randomised design (CRD) was used to evaluate four formulations with different pork-back fat-substitution levels: control formulation of emulsified alpaca-based sausages with 0.00% hydrated RDF-PP (C); emulsified alpaca-based sausages with 3.29% hydrated RDF-PP (F1); emulsified alpaca-based sausages with 6.57% hydrated RDF-PP; and emulsified alpaca-based sausages with 9.86% hydrated RDF-PP (F3). Data were analysed using the one-way ANOVA procedure using the RStudio (v. 2023.09.1, Boston, MA, USA) software [40] with a significance level of 5% (α = 0.05). By verifying the significance (*p* < 0.05), the post hoc Tukey’s HSD test was applied to determine statistical differences among the formulations.

#### 2.7.3. Sensory Analysis

For sensory analysis, a non-parametric Mann–Whitney U test (*p* < 0.05) was used to compare the mean values of the best formulation of emulsified alpaca-based sausages against a C. Statistical analysis was performed with a significance level of 5% (α = 0.05) using STATISTICA and RStudio (v. 2023.09.1, Boston, MA, USA) [40] software.

## 3. Results and Discussion

### 3.1. Effect of RDF-PR and PS on Physicochemical Properties of RDF-PP

#### 3.1.1. Colour Measurement of RDF-PP

One of the important indicators of the quality of food products is colour [27]. Likewise, the colour of a product is closely related to consumer preferences [25,27]. As shown in Table 1, the effects of RDF-PR (HH:HU) and PS (µm) on colour parameters were assessed in terms of lightness (L*), redness (a*), yellowness (b*), chroma (C), and hue angle (h°). The response surface plots (Figure 2) showed that at finer PS (<177 µm), regardless of RDF-PR, the L*, a*, C, and h° values increased significantly (*p* < 0.05). Conversely, the b* values showed a significant (*p* < 0.05) decrease. On the one hand, the ANOVA (Table 2) showed that the RDF-PR factor had a significant (*p* < 0.05) linear effect on L* and h° and a highly significant (*p* < 0.01) quadratic effect on a*, b*, C, and h°. On the contrary, the linear and quadratic effects of the PS factor revealed an extremely (*p* < 0.0001) and very (*p* < 0.001) significant effect on L*, a*, b*, C, and h°, respectively. Thus, indicating that the PS factor has a greater influence on the colourimetric characteristics of RDF-PP. Furthermore, the low values of predicted residual error sum of squares (PRESS; 1.99–18.7) and adequate precision (Adeq. Prec.> 4.00) of L*, a*, b*, C, and h° revealed a suitable predictive quality of the model [39].

Mai et al. [25] also found a significant increase in L* and a* values and a significant decrease in b* when reducing PS from 595 to 105 µm in HU peels. This contrast is probably due to the increase in surface area, which enhances light scattering by exposing the internal structure of cellulose and hemicellulose [41,42,43]. Accordingly, the higher the a* value and the lower the b* value, the higher the C and h° values. Hence, a pinkish luminous RDF-PP (Figure 1) was obtained for T1, T4, and T7 at the lowest levels (<177 µm) of PS (Table 1). Our study highlights the importance of the RDF-PP colour from both HH and HU species as an important parameter for the development of brighter food products, such as emulsified sausages.

#### 3.1.2. Techno-Functional and Physicochemical Properties of RDF-PP

The effects of RDF-PR and PS factors on the techno-functional (WHC, OHC, and SC) and physicochemical (PY, CF, and DE) properties of RDF-PP are shown in Table 3. Regarding WHC, the ANOVA (Table 2) showed an extremely significant (*p* < 0.0001) interaction between RDF-PR and PS, with a low PRESS (19.4) and a high Adeq. Prec. (36.3). However, the response surface plot (Figure 3A) showed a different trend as PS was reduced at the RDF-PR levels of HH(0%):HU(100%) and HH(100%):HU(0%). The main effects of RDF-PR and PS on WHC are shown in Appendix A. For instance, when 100% of HU peels were used, the highest WHC (31.9 g/g peels) was obtained at the lowest PS level (<177 µm), whereas 100% HH peels exhibited the lowest WHC (17.3 g/g peels). On the one hand, Zhuang et al. [29] reported a high WHC (54.20 g/g) in HU peels when reducing the PS level from 178 to 58 µm; on the other hand, Mai et al. [25] reported a low WHC (10.11 g/g) in HU peels when reducing the PS from 595 to 105 µm. Although WHC is influenced by starch, damaged starch, protein, capillary pore size, and capillary distribution [44,45], it is also associated with hemicellulose and pectin [46]. In this study, a significant Pearson correlation coefficient (*r* = 0.6022, *p* = 0.001) between WHC and PY was observed (Appendix A), indicating that WHC is notably influenced by the presence of uronic acid groups [47]. Considering that pectin is part of soluble fibre, Martínez-Girón et al. [48] supported this correlation by reporting an increase in water absorption capacity in red and yellow peach palm peels when reducing PS from 0.25 to 0.105 mm. According to Zhuang et al. [29] and Mai et al. [25], the finer the particle size, the better the water interaction with hydroxyl (–OH) groups [25]. Therefore, the divergence of WHC in HH(0%):HU(100%) and HH(100%):HU(0%) may be due to the pectin composition of *Hylocereus* species. In our research, T1 showed the highest WHC (31.9 g/g peels) using 100% HU peels at the lowest PS level (<177 µm) (Table 3), which revealed its potential to be used in meat product development as a fat replacer, juiciness retainer, and cooking loss reducer [22,49,50]. Nonetheless, Zhuang et al. [29] suggested that excessive reduction in RDF-PP PS (<58 µm) should be avoided because it may exhibit negative effects.

Regarding SC, the results of the ANOVA (Table 2) showed an extremely significant (*p* < 0.0001) interaction between RDF-PR and PS. Similarly, it demonstrated the suitable predictive quality of the model, as indicated by the low PRESS (29.3) and Adeq. Prec. > 4.00. Notably, the use of 100% HU peels exhibited the highest SC (46.9 mL H_2_O/g peels) at a PS <177 µm (Table 3). The response surface plot (Figure 3B) is consistent with this result, suggesting that finer PS levels may substantially enhance the SC of various RDFP species, as previously reported by Zhuang et al. [29].

On the one hand, according to Tejada-Ortigoza et al. [47] and Zlatanović et al. [49], SC is directly associated with the cellulose component of fibres. However, our study showed a negative correlation (*r* = −0.6273, *p* < 0.05) between SC and CF (Appendix A). On the other hand, Iuga and Mironeasa [51] reported that pectin, a soluble fibre component, is an influencing factor for SC because of its higher WHC. Similarly, Larrosa and Otero [52] stated that peels of fruits have higher SC because of their soluble fibre content. These perspectives are supported by Zhuang et al. [29], who associated the increase in SC with the soluble fibre content when reducing the PS of HU peels from 178 to 58 µm.

In addition to its hydration properties, RDF-PP exhibited the ability to absorb oil, which is related to the polysaccharide structure of RDFP. The ANOVA results (Table 2) showed that the linear (*p* < 0.001) and quadratic (*p* < 0.05) effects of RDF-PR, as well as the linear effect (*p* < 0.0001) of PS, substantially decreased the PRESS (0.04), indicating the suitable predictive performance of the regression model. Furthermore, the high Adeq. Prec. value (47.0) indicated the suitable quality of the regression model. Moreover, considering that the interaction between PS and RDF-PR was not significant (*p* > 0.05) (Table 2), the response surface plot (Figure 3C) suggests that only PS is the determining factor to enhance OHC. Therefore, it was shown that finer PS (<177 µm) significantly (*p* < 0.05) increased the OHC in T1 (2.11 g oil/g peels), T4 (2.13 g oil/g peels), and T7 (2.04 g oil/g peels) at all RDF-PR (Table 3). The main effects of RDF-PR and PS on OHC are shown in Appendix A. Similar results were obtained by Mai et al. [25] and Zhuang et al. [29] when PS was reduced in HU peels. According to Ahmed et al. [41], Martínez-Girón et al. [48], Tejada-Ortigoza et al. [47], and Zlatanović et al. [49], an increase in OHC is attributed to the structural composition, surface properties, and overall charge density of fibres. In our study, the observed increase in OHC at the finest PS may be attributed to the occurring capillary attraction facilitated by the physical retention of oil in honeycomb-like compartments of RDF-PP resulting from the milling process [25,48]. Additionally, it is likely that the reduction in PS may have led to a decrease in the number of hydroxyl groups in RDF-PP, consequently enhancing the hydrophobic capacity of the food material [25]. An adequate OHC is vital for flavour retention and yield, especially in cooked meat products prone to fat loss during cooking [53].

Pectin is a polysaccharide rich in galacturonic acid that stands out as a significant constituent within soluble fibres [33]. As shown in Table 2, the ANOVA for PY showed an extremely significant (*p* < 0.0001) interaction between PS and RDF-PR. Likewise, although the PRESS value was relatively high at 50.9, the Adeq. Prec. (25.4) still indicated the suitable quality of the model. Table 3 shows that T1 had the highest PY (27.3%) at a PS < 177 µm when using 100% HU peels, which is in accordance with Mai et al. [25]. In contrast, the use of 100% HH peels revealed that T7 showed the lowest PY (13.7%) at a PS < 177 µm. The response surface plot (Figure 3D) demonstrated that as HH peel concentration, i.e., HH(100%):HU(0%), increases, PY decreases significantly. According to this, it is likely that the PY of HH peels negatively influenced the PY of HU peels, as exhibited in RDF-PR of HH(50%):HU(50%) at finer PS (<177 µm). The main effects of RDF-PR and PS on PY are shown in Appendix A. Considering that pectin is part of soluble fibre, it has been reported that excessive reduction in PS (<58 µm) decreased the soluble fibre content of HU peels [29]. According to Chen et al. [54], it is likely that mechanical forces, such as milling, disrupted the linear or branched regions of the pectin structure in 100% HH peels, leading to the breakdown of glycosidic bonds. However, the effect of PS reduction on the microstructure of RDFP tissue depends on the botanical origin because of differences in composition and resistance to mechanical forces [55].

Regarding CF, ANOVA (Table 2) showed an extremely significant (*p* < 0.0001) interaction between RDF-PR and PS factors. In addition, the low PRESS value (12.9) indicated the suitable predictive performance of the regression model. However, the Adeq. Prec. value (10.0) was slightly greater than 4.00, indicating a variability in the experimental points around the regression line of the model. This may be explained by the non-significant (*p* > 0.05) variation in CF values (17.8–18.9%), as shown in T1, T2, T3, T5, T6, T8, and T9 (Table 3). On the other hand, the response surface plot (Figure 3E) showed that 100% of HU peels preserved their CF content as PS decreased, whereas 100% of HH peels underwent significant CF loss as PS decreased. Accordingly, HH(50%):HU(50%) showed a steady significant (*p* < 0.05) decrease as PS decreased. According to Parrott and Thrall [56], this may be due to the structural weakness of hemicellulose in response to mechanical forces, such as milling. This was confirmed by Zhuang et al. [29] and Mai et al. [25], who showed significant losses of the insoluble fibre of RDF-PP (HU) when PS was reduced to <58 µm and 105 µm, respectively.

The chemical structure of RDF-PP pectin treated under different RDF-PR and PS parameters was analysed by FT-IR spectroscopy. The FT-IR spectra of the pectin samples are shown in Figure 4. The absorption peaks at 1745 cm^−1^ ($\upsilon_{\left(C=O\right)COOMe}$) and 1630 cm^−1^ ($\upsilon_{\bar{as}(COO-)}$) were assigned to the C=O stretching vibration of the esterified carboxylic groups and the asymmetrical stretching vibrations of the carboxylate anions (COO^−^) groups, respectively [57,58]. According to Muhammad et al. [33], the high DE may be determined based on a stronger absorption at 1745 cm^−1^ coupled with a weaker absorption at 1630 cm^−1^. Therefore, T1 and T8 showed the highest DE. The peak areas at 1745 cm^−1^ ($\upsilon_{\left(C=O\right)COOMe}$) and 1630 cm^−1^ ($\upsilon_{\bar{as}(COO-)}$) in the FT-IR spectrum were used to determine the degree of ethoxylation [56].

DE measures the percentage of galacturonic acid units in the pectin structure that are esterified into the methoxyl group at C-6 and the acetyl group at C-2 and/or C-3 [59]. As shown in Table 2, the ANOVA showed that the PS factor had a highly significant (*p* < 0.01) linear and quadratic effect, whereas the RDFF-PR factor showed a non-significant (*p* > 0.05) linear effect, thus indicating the greater influence of PS on DE. However, the PRESS (37.4) and Adeq. Prec. (>4.00) values exhibited the poor predictive quality of the model, probably due to the lack of significance among most treatments (Table 3).

#### 3.1.3. Statistical Validation of Optimal Treatment of RDF-PP

Fibre-rich products used in food formulations involve the study of their techno-functional properties, such as WHC and OHC [51]. Fibres with high WHC can be used in meat product development for fat substitution, juiciness maintenance, and cooking loss reduction [22,49,50]. Likewise, OHC is vital for flavour retention and yield, especially in cooked meat products prone to fat loss during cooking [53]. On the other hand, pectin is a gel-forming agent with fat-substitution capacity because of its functional characteristics, such as gelling and texture properties in meat products [17]. T1 revealed the highest WHC, OHC, and PY at a RDF-PR of 100% HU and PS < 177 µm (Table 3). Therefore, the response variables WHC, OHC, and PY were chosen and replicated for validation, as shown in Table 4. The results showed that the observed values in WHC, OHC, and PY were consistent (*p* > 0.05) with the predicted values derived from the response surface model (Figure 3). Thus, T1 was selected as the fibre-rich product to be used in the preparation of emulsified alpaca-based sausages.

The particle size distribution of RDF-PP from T1 (RDF-PR, HU at 100%; PS < 177 µm) is shown in Figure 5. The fine fraction showed an average PS (X_PS_) of 70 µm with an equivalent diameter at a cumulative frequency of 50% (D_50_) of 56 µm and an equivalent diameter at a cumulative frequency of 90% (D_90_) of 124 µm. Figure 5C shows that the fine fraction exhibited reddish particles from the exocarp fraction of RDFP. Nevertheless, white and translucent particles were also observed, which probably indicates the occurrence of particles from the mesocarp fraction.

### 3.2. Effect of Fat Replacement by Hydrated RDF-PP in Emulsified Alpaca-Based Sausages

Red dragon fruit peel powder (RDF-PP) from T1 (Appendix A) was used to develop the four formulations of emulsified alpaca-based sausages. The pork-back fat substitution by hydrated RDF-PP caused a decrease of 15.0% from the C to 5.14% from F3. This maximum reduction was compensated using 9.86% hydrated RDF-PP, which comprised 0.30% dehydrated RDF-PP and 9.56% water (Appendix A). The amount of water used for each meat formulation was based on the WHC (31.9 g/g dehydrated peels) of RDF-PP, as shown in Table 3.

#### 3.2.1. Colour Measurement of Emulsified Alpaca-Based Sausages

As shown in Table 5, pork-back fat replacement by hydrated RDF-PP significantly (*p* < 0.05) influenced the colourimetric properties of alpaca-based sausages. Based on the L* (56.3–48.9) and b* (13.0–12.4) values, a significant decrease was observed as fat substitution increased from 0.00 to 9.86% hydrated RDF-PP. In contrast, the a* coordinate showed a significant increase (17.2–19.3) as more hydrated RDF-PP was added. Accordingly, C and h° were noticeably influenced as a* and b* varied. Considering that a* was higher than b*, C increased from 19.2 to 22.9. On the contrary, h° showed a significant decrease from 36.7 to 32.6°. These colourimetric changes led to the development of redder alpaca-based sausages, as previously shown by Aukkanit et al. [23]. Alves et al. [2] reported that Bologna-type sausages showed an L* decrease from 58.8 to 54.7 when substituting 20, 40, 60, 80, and 100% pork-back fat with a mixture of pork skin, green banana flour, and water. On the other hand, the increase in a* value in the alpaca-based sausages is probably due to the higher concentration of betalains as the RDF-PP concentration increases [60]. Given these results, among the formulations tested, as shown in Table 5, F3 (9.86% fat replacement) showed the lowest lightness (*L) and yellowness (b*) and the highest level of redness (a*).

#### 3.2.2. Physicochemical Characteristics of Emulsified Alpaca-Based Sausages

For moisture content, a significant (*p* < 0.05) increase was observed as more hydrated RDF-PP was added to the meat emulsion. Consequently, F3 showed the highest moisture content (73.2%). This result agrees with that of Aukkanit et al. [23], who reported an increase in moisture content from 61.2 to 63.4% in emulsion sausages when substituting 0.50, 1.00, 1.50, 2.00, and 2.50% pork-back fat with RDFP. These results revealed that the use of fibre-rich by-products, such as RDFP, could confer more juiciness on meat products, such as emulsified sausages. This result also indicated a higher water activity (a_w_ > 0.98) in all formulations but with no significant (*p* > 0.05) difference among them.

As more hydrated RDF-PP was added to the formulation, a significant (*p* < 0.05) decrease in the protein content (from 15.9 to 15.3%) in alpaca-based sausages was observed, probably due to a higher dilution of the meat proteins. In other words, the higher the hydrated RDF-PP levels (3.29 (F1), 6.57 (F2), and 9.86% (F3)), the greater the water quantity in the meat formulation, which would explain the lower protein content. In a study, Aukkanit et al. [23] reported even higher protein losses from 14.1 to 10.7% when replacing pork-back fat with dragon fruit peel powder in emulsified sausages. Based on the findings by Alves et al. [2], pork-back fat substitutes, such as water, starches, gums, and different fibre-rich raw materials, can reduce the protein content, depending on the fat replacement level.

As shown in Table 5, pork-back fat substitution by hydrated RDF-PP significantly (*p* < 0.05) decreased the fat content of emulsified alpaca-based sausages. According to Aukkanit et al. [23], the fat content of emulsified sausages significantly decreased after replacing pork-back fat with dragon fruit (HU) peels by up to 2.50%. Similarly, Alves et al. [2] reported a significant decrease in the fat content of Bologna-type sausages when using an emulsion constituted of pork skin, green banana flour, and water, up to 100% pork-back fat substitution. According to Regulation (EC) N° 1924/2006 [61], a claim stating that the content of one or more nutrients has been reduced may only be made where the reduction in content is at least 30% compared to a similar product. In this context, F3 meets the previous statement and exhibits the highest fat reduction (47.3%) compared to the C. Therefore, F3 meets the criteria for being classified as a ‘reduced-fat’ product.

#### 3.2.3. Frying Loss and Cooking Yield

Frying loss (FL) measures the ability of an emulsion to bind water, fat, and other ingredients during frying [3]. The substitution of pork-back fat with hydrated RDF-PP showed a significant (*p* < 0.05) effect on cooking yield (CY) and FL (Table 5). Specifically, the higher the pork-back fat substitution by hydrated RDF-PP, the lower the CY (72.2–66.9%) and the higher the FL (27.8–33.0%). This finding is in accordance with Vilcapoma et al. [20], who reported significant losses (28.0–30.6%) by substituting pork-back fat with yellow dragon fruit peel powder. In our research, the addition of hydrated RDF-PP revealed that F3 exhibited the highest FL at 33.0% and the lowest CY at 66.9%, indicating that the higher the pork-back fat substitution, the greater the addition of dehydrated RDF-PP and water. Consequently, this influenced the meat emulsion stability [62].

#### 3.2.4. Texture Characteristics of Emulsified Alpaca-Based Sausages

The texture characteristics of emulsified alpaca-based sausages are shown in Table 5. The hardness and chewiness properties were significantly (*p* < 0.05) influenced by the amount of hydrated RDF-PP added to the alpaca-based formulations (F1, F2, and F3). According to Alves et al. [2], when substituting pork-back fat with a mixture of pork skin, water, and green banana flour, hardness (from 84.9 to 61.2 N) and chewiness (from 54.9 to 32.9 N) losses occurred in Bologna-type sausages, probably due to the high WHC of fat substitutes. By chemically retaining water through the pectin matrix and swelling on contact with water, the hydrated RDF-PP reduced the hardness (from 49.9 to 34.8 N) and chewiness (from 26.9 to 21.7 N) of alpaca-based sausages. These findings showed higher values than those reported by Vilcapoma et al. [20], who substituted pork-back fat up to 78% in alpaca-based sausages. Likewise, the authors obtained reduced hardness and chewiness values of 16.2 N and 6.02 N, respectively [20].

Regarding cohesiveness, the addition of hydrated RDF-PP showed a non-significant (*p* > 0.05) increase from 0.606 to 0.633 in alpaca-based sausages (Table 5). According to Zaini et al. [22], chicken sausages enriched with banana peel powder showed a cohesiveness decrease from 0.64 to 0.47, caused by a reduction in fat content from 9.18 to 4.58%, revealing that fat is an important component that strengthens molecular bonds inside meat products. In our research, the addition of hydrated RDF-PP maintained the structure and integrity, as well as the springiness, of alpaca-based sausages during chewing, compared to C.

On the other hand, the adhesive force was inversely proportional to the adhesiveness of alpaca-based sausages (Table 5). The non-significant (*p*>0.05) increase in adhesive force may be explained by the effect of hydrated RDF-PP addition on the emulsification ability of formulations, leading to liquid loss during compression in the TPA test. Likewise, the significant (*p* < 0.05) decrease in adhesiveness may be explained by the addition of hydrated RDF-PP, leading to a decrease in the protein content of alpaca-based sausages (Table 5). Since these exudates contain less proteins, alpaca-based sausages had a juicier and less sticky consistency. According to Pereira et al. [63], sausages should be characterised by a smooth and firm surface without adherence to touch. In our study, F3 presented alpaca-based sausages with favourable characteristics in terms of a* (redness), moisture content, fat content, hardness, springiness, adhesive force, and adhesiveness.

### 3.3. Effect of Fat Replacement by Hydrated RDF-PP on Sensory Properties of Emulsified Alpaca-Based Sausages

F3 showed the most adequate colourimetric, physicochemical, and textural characteristics among C, F1, and F2, and is hereafter denominated ‘OF3′ (Optimal Formulation 3). Accordingly, the sensory evaluation of ‘OF3′ was carried out against the C. As shown in Figure 6, pork-back fat substitution had a significant (*p* < 0.05) effect on the sensory properties of emulsified alpaca-based sausages. The Mann–Whitney U test (Appendix A) indicated that the flavour, texture, and overall acceptability of ‘OF3′ differed from those of C, based on the lower values of U. Likewise, the lower Z values revealed significant (*p* < 0.05) differences for the sensory properties of flavour, texture, and overall acceptability. Therefore, OF3 showed significantly higher sensory scores in terms of flavour (5.89), texture (6.12), and overall acceptability (6.27) than C, with the scores for flavour (5.07), texture (5.43), and overall acceptability (5.51) (Appendix A).

Although ‘OF3′ colour did not show a significant (*p* > 0.05) increase, it was slightly higher than the colour reported for the C. Alves et al. [2] reported a low score for colour (7.19–6.30) of Bologna-type sausages when replacing pork-back fat with a mixture of pork skin, green banana flour, and water. In our research, ‘OF3′ showed a better score than C, probably due to the bright reddish colour of RDF-PP. Cáceres et al. [64] reported that consumers associate emulsified meat products with bright pink colours.

Nonetheless, the addition of hydrated RDF-PP improved the texture of emulsified alpaca-based sausages in ‘OF3′. As previously explained, despite the high WHC of RDF-PP, the crumbliness of sausages was avoided because of the gel-forming properties of RDF-PP. Petersson et al. [62] stated that sausages with low crumbliness have suitable gel-forming properties. Furthermore, OF3 recorded 5.73% of total dietary fibre content, which is significantly higher than the total dietary fibre (1%) of sausages enriched with rye bran [62].

In the context of future research, further exploration should be performed on the study of the techno-functional, physicochemical, and textural properties based on pectin isolated from RDF-PP using an ultrasound-assisted extraction. The use of this green technology will allow the evaluation of its potential application in various products, such as hamburgers, bread, cookies, and noodles. Additionally, the sensory evaluation of different emulsified meat formulations enriched with fibre-rich RDF-PP is being conducted in our research group to investigate consumer perceptions using the Check-All That Apply methodology.

## 4. Conclusions

The PS of RDF-PP significantly influenced all colourimetric, physicochemical, and techno-functional characteristics. Specifically, the lower the PS, the higher the WHC and OHC, and the lower the PY. On the other hand, the RDF-PR had a lower impact on the physicochemical characteristics, mainly the colourimetric characteristics. Also, the higher the percentage of HU peels, the greater the WHC, OHC, and PY.

The low-methoxyl pectin found in RDF-PP established its potential use in low-acid meat products, such as emulsified sausages. The application of low-methoxyl pectin-rich RDF-PP in emulsified alpaca-based sausages allowed a pork-back fat reduction of 15.0% in the C to 5.14% in our developed reduced-fat sausage formulation. The highest fat decrease was offset by employing 9.86% hydrated RDF-PP, made up of 0.30% dehydrated RDF-PP and 9.56% water, using a WHC of 31.86 g/g dehydrated peels, previously determined.

The developed reduced-fat sausage showed a redder colour with reduced hardness and chewiness and higher FL, showing the highest-rated flavour, texture, and overall acceptability against a C without significantly affecting the appearance, colour, or odour. Overall, these results indicated that the use of fibre-rich agroindustry residues, such as RDF-PP, may be a promising fat substitute in emulsified meat products, provide healthier alternatives, and allow the development of reduced-fat meat products with improved sensory qualities.

## Figures and Tables

**Figure 1 foods-13-00386-f001:**
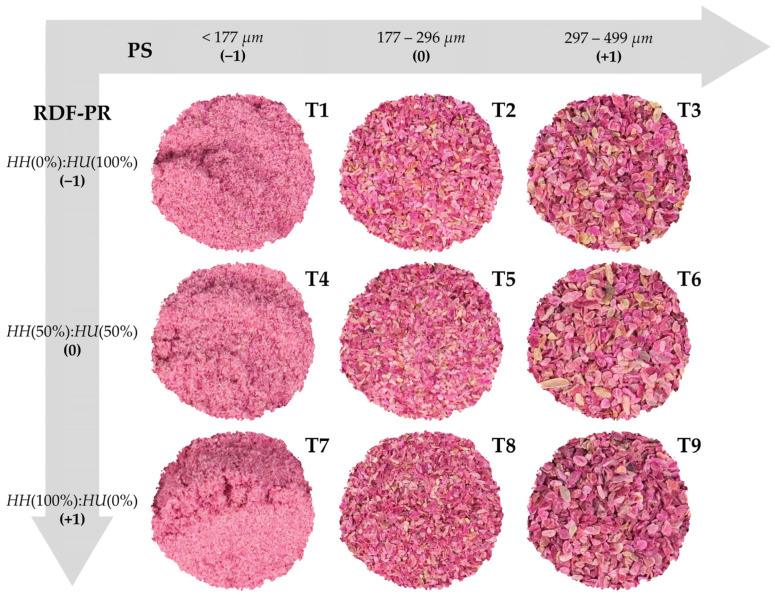
Experimental design illustrating the three-level parameters of the two variables: red dragon fruit peels ratio (RDF-PR; HH(0%):HU(100%), HH(50%):HU(50%), and HH(100%):HU(0%)) and particle size (PS; <177 µm, 177–296 µm, and 297–499 µm) based on a 3^k^ full factorial design.

**Figure 2 foods-13-00386-f002:**
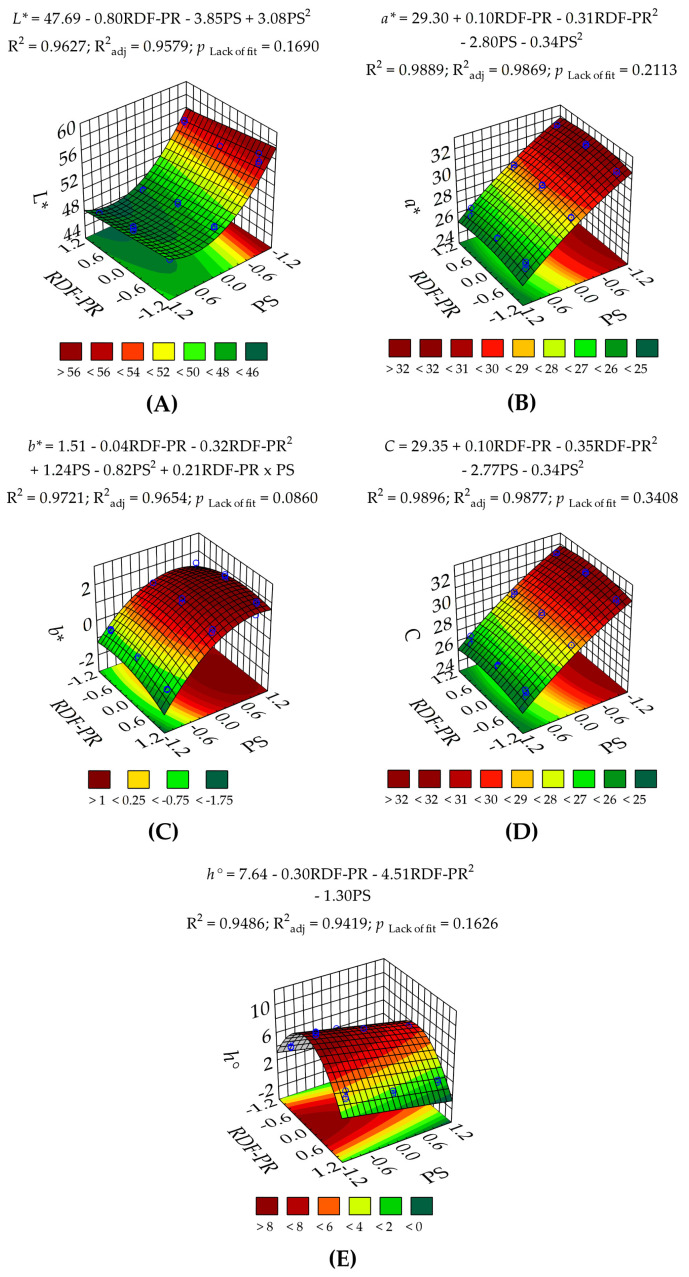
Three-dimensional (3D) response surface plots for colourimetric characteristics: L* (**A**), a* (**B**), b* (**C**), C (**D**), and h° (**E**) of red dragon fruit peel powder (RDF-PP) based on the 3^k^ full factorial design involving the independent variables: red dragon fruit peels ratio (RDF-PP; HH(0%):HU(100%): −1.0, HH(50%):HU(50%): 0.0, HH(100%):HU(0%): +1.0) and particle size (PS; <177 µm: −1.0, 177–296 µm: 0.0, and 297–499 µm: +1.0); where −1, 0, +1 denote coded values for RDF-PP or PS.

**Figure 3 foods-13-00386-f003:**
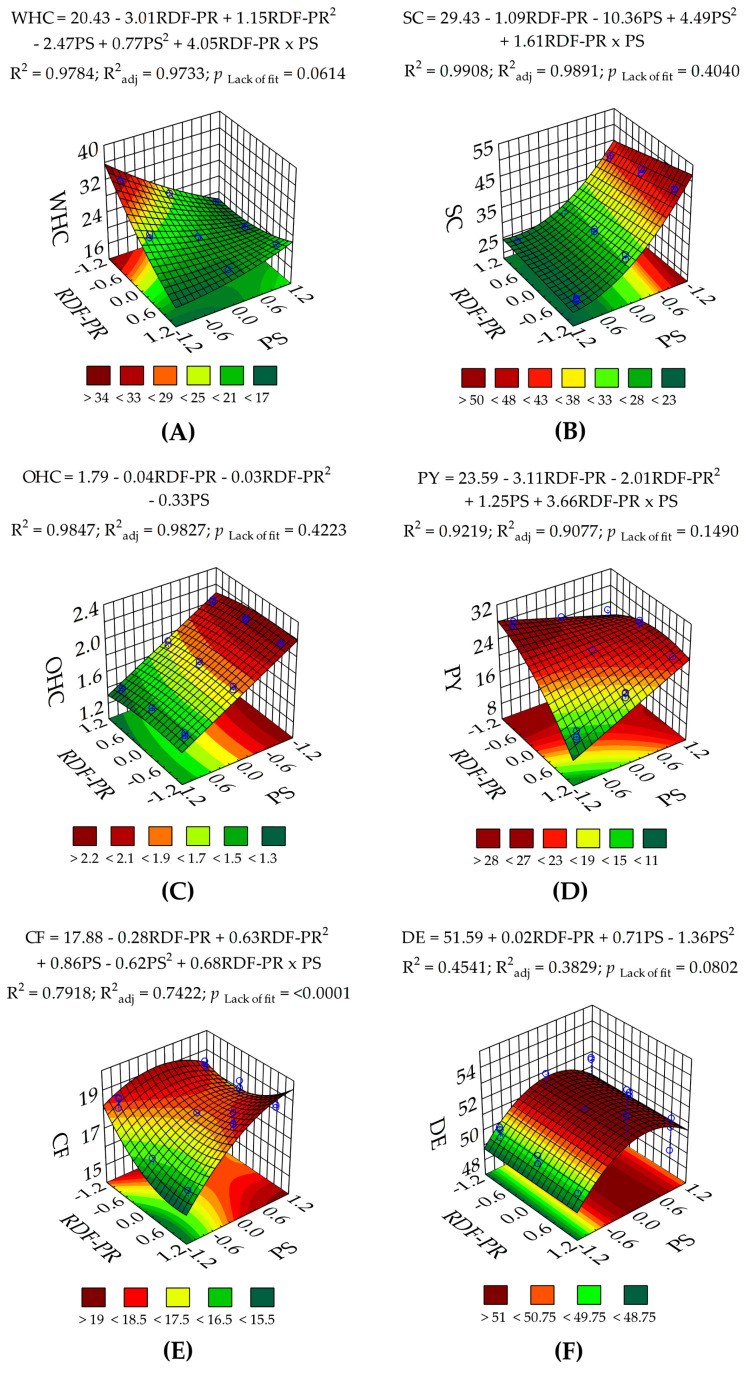
Three-dimensional (3D) response surface plots showing the effect of red dragon fruit peels ratio (RDF-PR; HH(0%):HU(100%): −1.0, HH(50%):HU(50%): 0.0, and HH(100%):HU(0%): +1.0) and particle size (PS; <177 µm: −1.0, 177–296 µm: 0.0, and 297–499 µm: +1.0) on the techno-functional (WHC (**A**), SC (**B**), and OHC (**C**)])and physicochemical (PY (**D**), CF (**E**), and DE (**F**)) properties of red dragon fruit peel powder (RDF-PP), where −1, 0, +1 denote coded values for RDF-PP or PS.

**Figure 4 foods-13-00386-f004:**
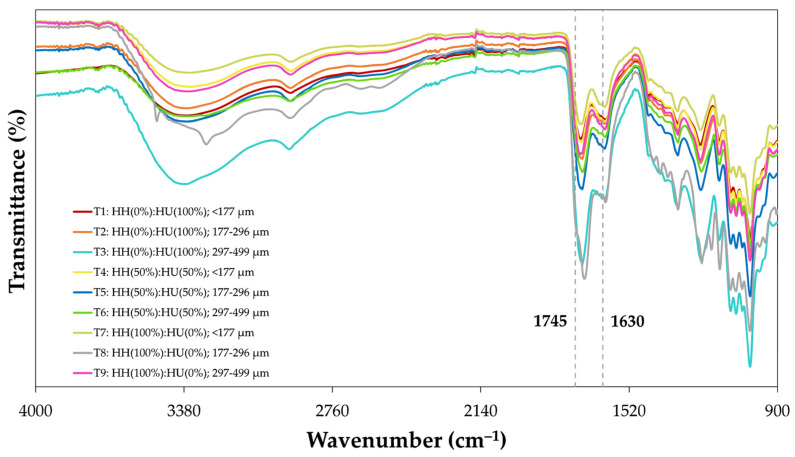
FT-IR spectra of pectin from red dragon fruit peel powder (RDF-PP) at different red dragon fruit peels ratio (RDF-PR; HH(0%):HU(100%), HH(50%):HU(50%), and HH(100%):HU(0%)) and particle size (PS; <177 µm, 177–296 µm, and 297–499 µm) based on nine treatments of the 3^k^ full factorial design (Appendix A).

**Figure 5 foods-13-00386-f005:**
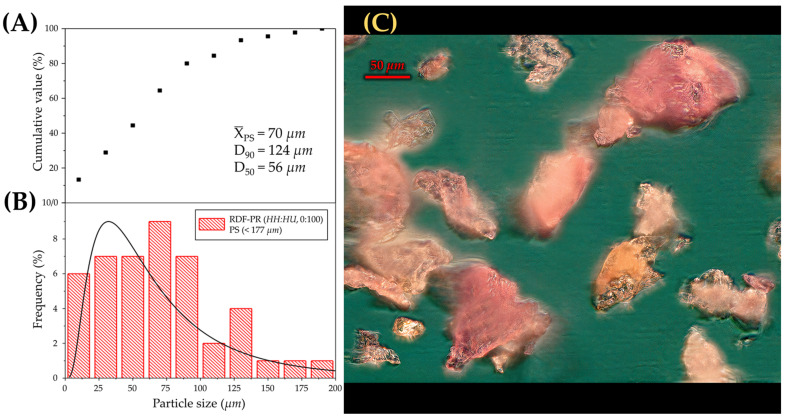
Particle size distribution by (**A**) cumulative value, (**B**) frequency, and (**C**) morphological features of red dragon fruit peel powder (RDF-PP) at the optimal red dragon fruit peels ratio (RDF-PR; HH(0%):HU(100%)) and particle size (PS; <177 µm).

**Figure 6 foods-13-00386-f006:**
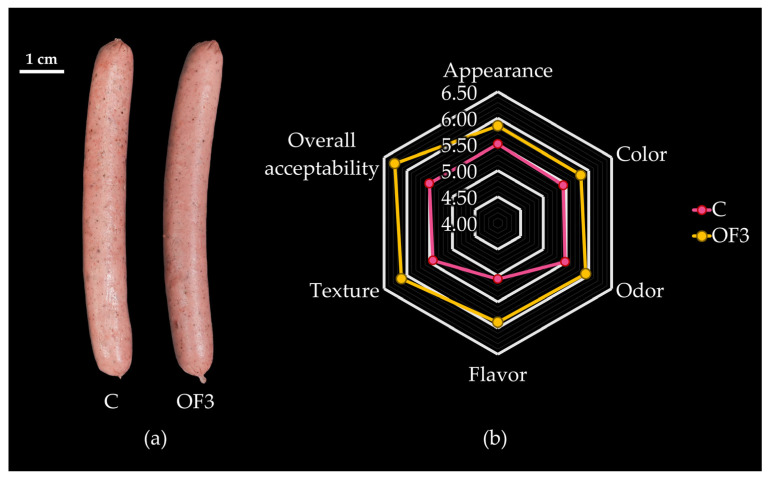
Sensory analysis of emulsified alpaca-based sausages: (**a**) morphological features of sausages with two different fat replacement levels of 0.00% (C, control formulation) and 9.86% (OF3, optimal formulation 3) hydrated red dragon fruit peel powder; and (**b**) sensory properties (appearance, colour, odour, flavour, texture, and overall acceptability) of C and OF3 sausages.

**Table 1 foods-13-00386-t001:** Colourimetric characteristics (L*, a*, b*, C, and h°) of red dragon fruit peel powder (RDF-PP) based on the 3^k^ full factorial design involving the independent variables: red dragon fruit peels ratio (RDF-PR; HH(0%):HU(100%), HH(50%):HU(50%), and HH(100%):HU(0%)) and particle size (PS; <177 µm, 177–296 µm, and 297–499 µm).

T	Uncoded and Coded Values				L*	a*	b*	C	h°
	RDF-PR (HH(%):HU(%))		PS (µm)						
T1	0:100	(−1)	<177	(−1)	55.8 ± 0.87 ^a^	31.3 ± 0.19 ^a^	−0.49 ± 0.05 ^d^	31.3 ± 0.18 ^a^	4.33 ± 0.42 ^c^
T2	0:100	(−1)	177–296	(0)	48.5 ± 0.17 ^b^	28.7 ± 0.19 ^c^	1.13 ± 0.17 ^c^	28.7 ± 0.18 ^c^	3.64 ± 0.99 ^cd^
T3	0:100	(−1)	297–499	(+1)	47.2 ± 0.36 ^bc^	25.9 ± 0.30 ^d^	1.40 ± 0.26 ^bc^	25.9 ± 0.30 ^d^	2.35 ± 0.75 ^de^
T4	50:50	(0)	<177	(−1)	54.0 ± 0.35 ^a^	31.8 ± 0.13 ^a^	−0.72 ± 0.07 ^de^	31.8 ± 0.13 ^a^	9.44 ± 0.27 ^a^
T5	50:50	(0)	177–296	(0)	47.9 ± 0.95 ^bc^	29.4 ± 0.14 ^b^	1.55 ± 0.14 ^abc^	29.5 ± 0.14 ^b^	7.44 ± 0.56 ^b^
T6	50:50	(0)	297–499	(+1)	47.6 ± 0.30 ^bc^	25.9 ± 0.10 ^d^	2.07 ± 0.15 ^a^	26.1 ± 0.09 ^d^	6.06 ± 0.31 ^b^
T7	100:0	(+1)	<177	(−1)	54.1 ± 1.50 ^a^	31.5 ± 0.21 ^a^	−1.06 ± 0.04 ^e^	31.5 ± 0.21 ^a^	4.28 ± 0.65 ^c^
T8	100:0	(+1)	177–296	(0)	46.7 ± 0.08 ^bc^	29.1 ± 0.19 ^bc^	1.21 ± 0.12 ^bc^	29.2 ± 0.19 ^bc^	2.39 ± 0.33 ^de^
T9	100:0	(+1)	297–499	(+1)	45.9 ± 0.24 ^c^	25.9 ± 0.54 ^d^	1.68 ± 0.41 ^ab^	25.9 ± 0.55 ^d^	1.83 ± 0.19 ^e^

Data are expressed as means ± standard deviation (*n =* 3). Mean values with different superscript letters within a column indicate significant differences (α = 0.05). T: treatment; L*: lightness (−); a*: green–red coordinate (−); b*: blue–yellow coordinate (−); C: chroma (−); h°: hue angle (−). The coded values (−1, 0, +1) for RDF-PR or PS are shown in parentheses.

**Table 2 foods-13-00386-t002:** Summary of the analysis of variance (ANOVA) and goodness-of-fit for the 3^k^ full factorial design showing F-test and *p*-values of linear (L) and quadratic (Q) models with interaction terms for the colourimetric characteristics (L*, a*, b*, C, and h°), techno-functional (WHC, OHC, and SC), and physicochemical (PY, CF, and DE) properties of red dragon fruit peel powder (RDF-PP).

Dependent Variables	ANOVA							Goodness-of-Fit					
	Model	RDF-PR (L)	RDF-PR (Q)	PS (L)	PS (Q)	RDF-PR × PS	Lack of Fit	C. V. (%)	PRESS (−)	R^2^	R^2^ Adj	R^2^ Pred	Adeq. Prec. (−)
L*	207 (<0.0001)	23.6 (0.00013)	—	553 (<0.0001)	118 (<0.0001)	—	1.77 (0.1690)	1.48	18.7	0.9627	0.9579	0.9462	28.4
a*	454 (<0.0001)	2.62 (0.1229)	9.10 (0.0074)	21.7 (<0.0001)	10.7 (0.0042)	—	1.62 (0.2113)	0.96	2.85	0.9889	0.9869	0.9802	42.7
b*	132 (<0.0001)	0.60 (0.4472)	16.6 (0.0007)	754 (<0.0001)	109 (<0.0001)	14.7 (0.0012)	2.57 (0.0860)	29.5	1.99	0.9720	0.9654	0.9409	25.2
C	484 (<0.0001)	2.58 (0.1252)	11.1 (0.0037)	2152 (<0.0001)	10.7 (0.0041)	—	1.21 (0.3408)	0.93	2.63	0.9896	0.9877	0.9814	44.3
h°	162 (<0.0001)	5.38 (0.0321)	395 (<0.0001)	98.8 (<0.0001)	—	—	1.80 (0.1626)	12.1	10.9	0.9486	0.9419	0.9322	30.0
WHC	180 (<0.0001)	414 (<0.0001)	19.9 (0.0002)	278 (<0.0001)	9.02 (0.0076)	497 (<0.0001)	2.93 (0.0614)	3.36	19.4	0.9784	0.9733	0.9604	36.3
OHC	555 (<0.0001)	19.6 (0.0003)	5.11 (0.0363)	1469 (<0.0001)	—	—	1.04 (0.4223)	1.96	0.04	0.9846	0.9826	0.9790	47.0
SC	589 (<0.0001)	24.3 (0.00011)	—	2200 (<0.0001)	137 (<0.0001)	35.3 (<0.0001)	1.06 (0.4040)	2.92	29.3	0.9908	0.9891	0.9862	50.9
PY	75.1 (<0.0001)	136 (<0.0001)	18.9 (0.0003)	22.2 (0.0002)	—	126 (<0.0001)	1.93 (0.1490)	5.10	50.9	0.9219	0.9077	0.8767	25.4
CF	14.6 (<0.0001)	12.7 (0.0022)	21.6 (0.0002)	121 (<0.0001)	20.6 (0.0002)	50.1 (<0.0001)	13.8 (0.00007)	3.28	12.9	0.7917	0.7421	0.5891	10.0
DE	6.47 (0.0028)	0.01 (0.9178)	—	11.1 (0.0037)	13.7 (0.0016)	—	2.37 (0.0802)	2.01	37.4	0.4540	0.3828	0.1110	5.72

RDF-PR: red dragon fruit peels ratio; PS: particle size; WHC: water-holding capacity (g/g peels); OHC: oil-holding capacity (g oil/g peels); SC: swelling capacity (mL H_2_O/g peels); PY: pectin yield (%); CF: crude fibre (%); DE: degree of esterification (%). ‘—‘: term dropped from the analysis due to non-significance (*α* = 0.05). Measurements of goodness-of-fit were as follows: coefficient of variation (C. V.), predicted residual error sum of squares (PRESS), coefficient of determination (R^2^), R^2^ adjusted (R^2^ adj.), R^2^ predicted (R^2^ pred.), and adequate precision (Adeq. Prec.).

**Table 3 foods-13-00386-t003:** Techno-functional (WHC, OHC, and SC) and physicochemical (PY, CF, and DE) properties of red dragon fruit peel powder (RDF-PP) from the interaction of red dragon fruit peels ratio (RDF-PR; HH(0%):HU(100%), HH(50%):HU(50%), and HH(100%):HU(0%)) and particle size (PS; <177 µm, 177–296 µm, and 297–499 µm) based on the 3^k^ full factorial design.

T	WHC	OHC	SC	PY	CF	DE
T1	31.9 ± 0.09 ^a^	2.11 ± 0.02 ^a^	46.9 ± 0.03 ^a^	27.3 ± 1.29 ^a^	18.4 ± 0.31 ^ab^	49.9 ± 0.38 ^b^
T2	24.3 ± 0.93 ^b^	1.81 ± 0.04 ^b^	30.1 ± 1.85 ^c^	25.2 ± 1.12 ^ab^	18.2 ± 0.27 ^ab^	50.8 ± 1.35 ^ab^
T3	19.1 ± 0.04 ^cd^	1.48 ± 0.03 ^c^	23.3 ± 0.25 ^d^	21.6 ± 1.98 ^cd^	18.5 ± 0.17 ^ab^	51.5 ± 1.02 ^ab^
T4	24.1 ± 1.45 ^b^	2.13 ± 0.01 ^a^	44.3 ± 1.07 ^a^	21.6 ± 0.10 ^cd^	16.2 ± 0.48 ^c^	49.5 ± 0.47 ^b^
T5	20.1 ± 0.43 ^cd^	1.80 ± 0.02 ^b^	30.2 ± 1.73 ^c^	23.4 ± 0.19 ^bc^	17.8 ± 0.23 ^b^	51.2 ± 0.31 ^ab^
T6	18.6 ± 0.34 ^de^	1.44 ± 0.04 ^cd^	23.0 ± 0.10 ^d^	25.8 ± 1.25 ^ab^	18.4 ± 0.45 ^ab^	51.0 ± 0.36 ^ab^
T7	17.3 ± 0.12 ^e^	2.04 ± 0.07 ^a^	41.5 ± 0.52 ^b^	13.7 ± 1.10 ^e^	15.9 ± 0.44 ^c^	49.1 ± 0.37 ^b^
T8	19.2 ± 0.09 ^cd^	1.75 ± 0.04 ^b^	28.0 ± 0.07 ^c^	19.1 ± 0.67 ^d^	18.9 ± 0.34 ^a^	52.8 ± 1.59 ^a^
T9	20.8 ± 0.48 ^c^	1.37 ± 0.03 ^d^	24.3 ± 0.04 ^d^	22.7 ± 1.16 ^bc^	18.7 ± 0.09 ^ab^	50.3 ± 1.08 ^ab^

Data are expressed as means ± standard deviation (*n =* 3). Mean values with different superscript letters within a column indicate significant differences (*p* < 0.05). T: treatment; WHC: water-holding capacity (g/g peels); OHC: oil-holding capacity (g oil/g peels); SC: swelling capacity (mL H_2_O/g peels); PY: pectin yield (%); CF: crude fibre (%); DE: degree of esterification (%).

**Table 4 foods-13-00386-t004:** Validated data according to water-holding capacity (WHC, g/g peel), oil-holding capacity (OHC, g oil/g peel), and pectin yield (PY, %) of red dragon fruit peel powder (RDF-PP) at a 95% confidence interval showing lower and upper limits.

Dependent Variables	Lower Limit	Upper Limit	Observed Value	Predicted Value
WHC	30.7	33.1	32.1 ^a^	31.9 ^a^
OHC	2.08	2.18	2.20 ^b^	2.13 ^b^
PY	25.3	28.9	27.1 ^c^	27.1 ^c^

Data are expressed as means ± standard deviation (*n =* 3). Mean values with different superscript letters in the same row indicate significant difference (α = 0.05).

**Table 5 foods-13-00386-t005:** Colourimetric (L*, a*, b*, C, and h°), physicochemical (a_w_, MC, FC, PC, CY, and FL) and texture profile (hardness, chewiness, cohesiveness, springiness, adhesiveness, and adhesive force) properties of emulsified alpaca-based sausages with different pork-back fat-substitution levels by the optimal treatment (T1) of hydrated red dragon fruit peel powder (RDF-PP).

Dependent Variables	Formulations			
	C	F1	F2	F3
Physicochemical properties				
L*	56.3 ± 0.05 ^a^	53.4 ± 0.02 ^b^	50.7 ± 0.34 ^c^	48.9 ± 0.43 ^d^
a*	17.2 ± 0.01 ^c^	17.3 ± 0.17 ^c^	18.4 ± 0.03 ^b^	19.3 ± 0.03 ^a^
b*	13.0 ± 0.05 ^a^	12.9 ± 0.04 ^b^	12.6 ± 0.01 ^c^	12.4 ± 0.01 ^d^
C	19.2 ± 0.04 ^d^	21.2 ± 0.01 ^c^	22.4 ± 0.03 ^b^	22.9 ± 0.02 ^a^
h°	36.7 ± 0.03 ^a^	36.3 ± 0.02 ^b^	35.3 ± 0.02 ^c^	32.6 ± 0.10 ^d^
a_w_	0.9817 ± 0.0000 ^a^	0.9842 ± 0.0000 ^a^	0.9825 ± 0.0000 ^a^	0.9814 ± 0.0000 ^a^
MC (%)	66.9 ± 0.10 ^d^	68.1 ± 0.53 ^c^	69.9 ± 0.04 ^b^	73.2 ± 0.27 ^a^
FC (%)	24.3 ± 0.45 ^a^	20.0 ± 0.84 ^b^	17.6 ± 0.88 ^c^	12.8 ± 0.42 ^d^
PC (%)	15.9 ± 0.13 ^a^	15.9 ± 0.09 ^a^	15.5 ± 0.15 ^b^	15.3 ± 0.12 ^b^
CY (%)	72.2 ± 1.22 ^a^	70.7 ± 0.77 ^ab^	69.9 ± 0.90 ^b^	66.9 ± 0.05 ^c^
FL (%)	27.8 ± 1.22 ^c^	29.3 ± 0.77 ^bc^	30.0 ± 0.90 ^b^	33.0 ± 0.05 ^a^
Texture properties				
Hardness (N)	49.9 ± 1.45 ^a^	41.3 ± 1.73 ^b^	40.0 ± 0.20 ^b^	34.8 ± 0.96 ^c^
Chewiness (N)	26.9 ± 0.78 ^a^	25.5 ± 1.98 ^ab^	22.9 ± 0.25 ^bc^	21.7 ± 0.24 ^c^
Cohesiveness (−)	0.606 ± 0.01 ^a^	0.616 ± 0.03 ^a^	0.633 ± 0.06 ^a^	0.633 ± 0.06 ^a^
Springiness (−)	0.867 ± 0.06 ^a^	0.900 ± 0.00 ^a^	0.867 ± 0.06 ^a^	0.900 ± 0.00 ^a^
Adhesiveness (mJ)	0.180 ± 0.02 ^a^	0.057 ± 0.02 ^b^	0.064 ± 0.01 ^b^	0.031 ± 0.01 ^b^
Adhesive force (N)	0.049 ± 0.001 ^a^	0.056 ± 0.008 ^a^	0.059 ± 0.021 ^a^	0.065 ± 0.005 ^a^

Data are expressed as means ± standard deviation (*n =* 3). Different letters in the same row indicate significant differences (α = 0.05). MC: moisture content; FC: fat content; PC: protein content; CY: cooking yield; FL: frying loss.

## Data Availability

The data presented in this study are available upon request from the corresponding author. The data are not publicly available due to privacy.

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
