# Peer review of "Red Dragon Fruit Peels: Effect of Two Species Ratio and Particle Size on Fibre Quality and Its Application in Reduced-Fat Alpaca-Based Sausages"

_foods, 2024, doi:10.3390/foods13030386_

Round 1

Reviewer 1 Report

Comments and Suggestions for Authors

Comments for Red Dragon Fruit Peels: Effect of the Proportion and Particle 2 Size on Fibre Quality and Its Application in Reduced-Fat Al- 3 paca-Based Sausagesis attractive and well structure research article, proper material and methods as properly discussed Result and discussion, for publication in journal “Foods “Overall, the manuscript is quite interesting, the content is well organized, the figures and the table summarize the main insights of the study. However, please enhance the language and address the following comments to improve the quality of this manuscript.

Revise the Line No. 20-21 from abstract section remove numbering like these (i)

2.     Revise the Lines No. 96- to Line No 102 by remove the numbering in introduction section like  (i)

3.     The Line should no start with “Next” revise the Line No. 136  

4.     Remove this “as previously described” from Line No. 141

5.     The Line should no start with “Using” Line No. 244

6.     Before Line 322 write the importance of Colorimetric Characteristics of RDF-PP for your study

7.     Write about  “techno-functional” and “physico-chemical’ propertiesbefore Line No. 352 why this these characteristics are important in the case of your study

8.     Remove H2O from this part (31.9 g H2O/g peels) of Line No. 358 and 358,  and anywhere this is present

9.       The Line No.601 is so short it’s connecting with previous Line No.  600

10.  Add the reference in Line No. 614

11.  Add full name of (FL) with this abbreviation in Line No. 592

12.  Revise the Line No. 629 it should not be start with word “As

13.  In conclusion please add further study required that will support this study , one should be Line after Line No. 691

Comments on the Quality of English Language

Please enhance the language.

Author Response

Response to Reviewer 1 Comments

Comments for Red Dragon Fruit Peels: Effect of the Proportion and Particle Size on Fibre Quality and Its Application in Reduced-Fat Alpaca-Based Sausagesis attractive and well structure research article, proper material and methods as properly discussed Result and discussion, for publication in journal “Foods“. Overall, the manuscript is quite interesting, the content is well organized, the figures and the table summarize the main insights of the study. However, please enhance the language and address the following comments to improve the quality of this manuscript.

Response: First, thank you very much for your appreciation of our manuscript by the reviewer. Second, thanks for your observation about the English grammar. We sent our manuscript to a native English speaker (English Editing and Manuscript Proofreading by EnagoTM) to improve the English grammar in our corrected manuscript, we are attaching the respective document. Finally, in “blue font”, you will find our answers to your suggestions/queries.

  1. Revise the Line No. 20-21 from abstract section remove numbering like these (i)

Response 1: Numbering ‘(i)’ removed in lines 20-21.

  1. Revise the Lines No. 96 to Line No 102 by remove the numbering in introduction section like (i)

Response 2: Thank you for your observation. Numbering ‘(i)’ removed in lines 113, 116, and 118.

  1. The Line should no start with “Next” revise the Line No. 136

Response 3: Done. The word “Next” has been removed in line 161.

  1. Remove this “as previously described” from Line No. 141

Response 4: Done. The phrase “as previously described” was deleted in line 166.

  1. The Line should no start with “Using” Line No. 244

Response 5: Thank you for your suggestion. The word “Using” has been removed and the sentence was restructured to: “A texture analyser (CTX, Brookfield Ametek, MA, USA) was used to compress each slice…” in lines 275 and 276.

  1. Before Line 322 write the importance of Colorimetric Characteristics of RDF-PP for your study

Response 6: Thank you for your contribution. The importance of the colourimetric characteristics of RDF-PP has been included in lines 378 – 380. Furthermore, additional information about the importance of colour as an indicator of the food quality has been included in lines 357 – 358.

  1. Write about “techno-functional” and “physico-chemical” properties before Line No. 352 why this these characteristics are important in the case of your study

Response 7: Thank you for your suggestion. In the study, three properties — dependent variables: water holding capacity (WHC), oil holding capacity (OHC), and pectin yield (PY) — of red dragon fruit peel powder (RDF-PP) were chosen in order to develop the emulsified alpaca-based sausages. Accordingly, the importance of each dependent variable was pointed out in the manuscript. Regarding the techno-functional properties, the importance of WHC is stated in lines 413 – 415. Regarding OHC, its importance is pointed out in lines 484 – 485. Regarding to the physicochemical properties, the importance of PY is indicated in lines 546 – 548. Furthermore, the section ‘3.1.3. Statistical Validation of Optimal Treatment of RDF-PP’ (line 541) indicate the importance of the properties chosen for the development of the meat product (lines 541 – 554).

  1. Remove H2O from this part (31.9 g H2O/g peels) of Line No. 358 and 358, and anywhere this is present

Response 8: Thank for your observation. Done. “H2O” was removed from the following parts of the manuscript: lines 26, 397, 398, 413, 429, 441, 555, 574, and 733. Currently, the term ‘g H2O/g peels’ has changed to ‘g/g peels’.

  1. The Line No.601 is so short it’s connecting with previous Line No. 600

Response 9: Done. The lines have been updated to lines 647 – 649.

  1. Add the reference in Line No. 614

Response 10: Thank you for your observation. The cite from ‘Vilcapoma et al.’, referenced as ‘[20]’, was allocated at the beginning of the sentence in line 645, which information ends in line 647.

  1. Add full name of (FL) with this abbreviation in Line No. 592

Response 11: Done. The full name of ‘FL’ has been added before ‘FL’. Therefore, the phrase ‘Frying loss (FL)’ has been included in line 641.

  1. Revise the Line No. 629 it should not be start with word “As”

Response 12: Done. The word “As” has been replaced by the adverb “Since” in line 678.

  1. In conclusion please add further study required that will support this study, one should be Line after Line No. 691

Response 13: Please, about this suggestion, we included the ‘further study’ at the end of the “Results and Discussion” section (lines 714 – 721), including this information: ‘In the context of future research, further exploration should be performed on the study of the techno-functional, physico-chemical, and textural properties based on pectin isolated from RDF-PP using an ultrasound-assisted extraction. The use of this green technology will allow the evaluation of its potential application in various products, such as hamburgers, bread, cookies, and noodles. Additionally, the sensory evaluation of different emulsified meat formulations enriched with fibre rich RDF-PP is being conducted in our research group to investigate consumer perceptions using the Check-All That Apply methodology’. If the reviewer agrees, we prefer to leave that information at the end of the above-mentioned section.

Reviewer 2 Report

Comments and Suggestions for Authors

This study selecting the species and particle size of red dragon fruit peels by their properties to used as fat substitution and then study on the application with alpaca-based sausages. Overall, the topic and scope of the work are interesting as they relate to the production of health-care foods that utilize by-products in food processing. The quality of MS is fair. While parts such as the introduction and conclusion are good, there are some areas require improvement (major). The abstract should be largely re-constructed and added more highlighted information. The experimental design should be clearly defined. Also, the discussion should be added in more detail.

Here is my comment:

Main issue:

- Topic: effect of proportion? Or species? Please check for the most suitable.

- Why did the author choose to study different species in the experimental design? What are the differences among these species? If differences existed, why didn't the author study them independently? This is because the experimental design might not provide distinct results for the factors of species and size.

ABSTRACT

- The abstract needs to be completely rewritten to enhance understanding and improve its clarity. Firstly, all experiments should be explicitly mentioned in the abstract (currently lacking texture or sensory evaluation details (the 3rd objective)). Furthermore, regarding the latest findings, the author mentioned that the 'T1' condition can replace 65.7%. What specific parameter did the author use as the standard for acceptability? Hence, I suggest incorporating empirical results, especially quantitative data, to reinforce the abstract.

INTRODUCTION

- Well-written. However, for improvement, I suggested to the author to condense the paragraph discussing the potential health risks associated with the high fat content of sausages. Instead, they could incorporate previous studies that highlight the properties of dragon peels or explore their potential applications as fat substitutes, not only for sausages but also for other related products.

MATERIAL AND METHODS

- Line105: Red dragon fruit from HH?? I suggested changing the word for better understanding. Also, how much to collect for this experiment? It seem like it brings from one place, thus, how to make sure they can be a good representative of each type?

- Line115: How to control the quality of peels before experiment? The transportation, duration after harvesting, or either storage conditions should be provided.

- Line120: What’s its final moisture content after drying? Please include. Moreover, how’s about the particle size experiment? At now, it seem likes all peels were milled with the same conditions????

- Interestingly, how come of these particle size of the peels? Why author designed by using these 3 sizes? Also, when the author experimented with particle size, what type of red dragon fruit was utilized? (Considering there are two types of this fruit.) If the author used only one type for the particle size experiment, how can they ensure that the outcomes would be similar for the other type?

- Line124, 225: Change the subtopic into “Color measurement”

- Line133: Change the subtopic into "Crude fiber content”

- Line206: The description of sausage production only covered the process without mentioning the substitution using the peels. Please include details about the experimental sets and the procedure for substituting the peels in the product.

- The four formulations of sausages are still confused.

-Line250: The preparation of sausage before allowing to assess for the scores should be provided. For example, re-heat? What’s size for serving? How’s many samples? What’s time duration during each sample?, etc.

- The statistical analysis section appears excessively long. I recommended that the author condense it by focusing on the main method used.

RESULTS AND DISCUSSION

- Which of the two species of red dragon fruit peels exhibits better properties? Furthermore, how does the particle size influence these properties? The author should delve into these details as I believe that utilizing an experimental design like this might not allow for the independent separation of each factor's effect.

- The author solely presented the results and compared them with other studies but failed to discuss the underlying reasons affected by the experimental factors. For instance, if 'T1' exhibits the best techno-functional properties, what is the reason behind it? Does it depend on the type of dragon peels or their size? Moreover, there seems to be a lack of specific properties data for individual dragon fruit peel species. Consequently, the results do not conclusively identify their precise origin or characteristics.

CONCLUSION

- Good.

Author Response

Response to Reviewer 2 Comments

This study selecting the species and particle size of red dragon fruit peels by their properties to used as fat substitution and then study on the application with alpaca-based sausages. Overall, the topic and scope of the work are interesting as they relate to the production of health-care foods that utilize by-products in food processing. The quality of MS is fair. While parts such as the introduction and conclusion are good, there are some areas require improvement (major). The abstract should be largely re-constructed and added more highlighted information. The experimental design should be clearly defined. Also, the discussion should be added in more detail.

Response: First of all, thank you very much for your appreciation of our manuscript by the reviewer. In “blue font”, you will find our answers to your major suggestions/queries.

Here is my comment:

Main issue:

- Topic: effect of proportion? Or species? Please check for the most suitable.

Response: Thank you for your suggestion. The title has been reviewed and the term ‘proportion’ was changed and improved to ‘… Two Species Ratio…’. Therefore, one additional word has been included in the final title: ‘Red Dragon Fruit Peels: Effect of Two Species Ratio and Particle Size on Fibre Quality and Its Application in Reduced-Fat Alpaca-Based Sausages’ (22 words), as compared to the original one (21 words). Furthermore, the term ‘proportion’ was updated to ‘ratio’ in line 105.

- Why did the author choose to study different species in the experimental design? What are the differences among these species? If differences existed, why didn't the author study them independently? This is because the experimental design might not provide distinct results for the factors of species and size.

Response: Thank for your valuable contribution. Both Hylocereus undatus and Hylocereus hybridum species are utilized in beverage industries for juice production in Peru. Likewise, both species are popular and exported to the European market. However, scarce research is performed regarding the reutilization of the dragon fruit peels waste. In the context of a circular economy, a practical procedure for taking advantage of red dragon fruit peels is to process both species residues without separating them.

Regarding the differences between species, we realized that the thickness of Hylocereus hybridum peels was higher than Hylocereus undatus peels, which would probably reveal a difference in pectin yield. Therefore, we had the following initial hypothesis: is there a difference in peel thickness between from Hylocereus undatus and Hylocereus hybridum species? Following this, the peel thickness, among other characteristics, was obtained through the pertinent measurement techniques. Table S1 from Supplementary materials shows the differences between both species. Subsequently, the pectin yield of peels from both species was independently determined. The results of the preliminary experiments showed that both Hylocereus hybridum and undatus peels had similar pectin yields (approximately 21.5 – 22.5 %), without considering “particle size” as an independent variable (factor), which inspired us to investigate the colourimetric, physicochemical, and techno-functional properties of different red dragon fruit peels ratio (RDF-PR).

On the other hand, we showed interest to investigate the effect of particle size because of its effect on the physicochemical, technofunctional, and colourimetric characteristics of red dragon fruit peels, as shown in this previous study: https://doi.org/10.1111/j.1365-2621.2012.02971.x. We observed that the effect of particle size influenced the above-mentioned characteristics of red dragon fruit peels, such as the water holding capacity, oil holding capacity, swelling capacity, and pectin yield. Accordingly, particle size, along with RDF-PR, was considered as the second independent variable (factor) in our study.

ABSTRACT

- The abstract needs to be completely rewritten to enhance understanding and improve its clarity. Firstly, all experiments should be explicitly mentioned in the abstract (currently lacking texture or sensory evaluation details (the 3rd objective). Furthermore, regarding the latest findings, the author mentioned that the 'T1' condition can replace 65.7%. What specific parameter did the author use as the standard for acceptability? Hence, I suggest incorporating empirical results, especially quantitative data, to reinforce the abstract.

Response: Firstly, we adjusted the original abstract based on the words limit. However, we consider your observation as a useful scope, which permits us to expand the scientific information regarding the original abstract. Regarding the experimental methodology, lines 29 – 31 show additional detail about the texture analysis and sensory evaluation performed on the emulsified alpaca-based sausages. Furthermore, additional details were included regarding the colourimetric, physicochemical, and texture results of the alpaca-based sausages in lines 33 – 37.

INTRODUCTION

- Well-written. However, for improvement, I suggested to the author to condense the paragraph discussing the potential health risks associated with the high fat content of sausages. Instead, they could incorporate previous studies that highlight the properties of dragon peels or explore their potential applications as fat substitutes, not only for sausages but also for other related products.

Response: Thank you very much for your suggestion. Regarding the paragraph about the potential health risks associated with the high fat content of sausages, the information was reviewed and summarized. The updated information is in lines 52 – 63. With respect to the incorporation of previous studies about red dragon fruit peels properties or applications, additional information has been added in lines 93 – 103. Some references utilized in the manuscript were included. On the other hand, the following new reference was added to the manuscript:

  • Utpott, M.; Ramos De Araujo, R.; Galarza Vargas, C.; Nunes Paiva, A.R.; Tischer, B.; De Oliveira Rios, A.; Hickmann Flôres, S. Characterization and Application of Red Pitaya (Hylocereus Polyrhizus) Peel Powder as a Fat Replacer in Ice Cream. J Food Process Preserv 2020, 44, doi:10.1111/jfpp.14420.

MATERIAL AND METHODS

- Line105: Red dragon fruit from HH?? I suggested changing the word for better understanding. Also, how much to collect for this experiment? It seem like it brings from one place, thus, how to make sure they can be a good representative of each type?

Response: Done. The introductory phrase “Red dragon fruit from HH” has been changed to “Red dragon fruit from Hylocereus hybridum (HH; …)” in line 123. Likewise, the term “and HU” was changed to “and Hylocereus undatus” in line 124. Regarding to the quantity of red dragon fruits collected, the quantities for each Hylocereus undatus and Hylocereus hybridum species were ⁓55 kg (updated in lines 123 and 124). Furthermore, both species were collected from the Corporation Abregú (Latitude: −11.41694; Longitude: −77.23363) harvested at the third flowering (lines 127 – 128) on May 03. Previously, a representative quantity (⁓3 kg) of each species was requested from the farmer in order to analyse them. Subsequently, the obtained results (fruit weight, peel thickness, soluble solids, acidity, moisture, yield, pH), in Table S1 from Supplementary materials, were corroborated with the results from the greater quantity (⁓55 kg) of red dragon fruits.

- Line115: How to control the quality of peels before experiment? The transportation, duration after harvesting, or either storage conditions should be provided.

Response: The collection of red dragon fruits was carried out with the advice of the farmer, who is a specialist in dragon fruit cultivation from the Corporation Abregú (https://pitahayaabregu.com/fundo/). Red dragon fruits from Hylocereus undatus and Hylocereus hybridum were harvested at an early stage (green and red colour) two days before transportation. In the first day, fruits from both species were placed in harvest containers and stored in a dark place at room temperature. In the second day, fruits were transferred to perforated cardboard boxes, sealed, placed in a dark place at room temperature, and transported by car from Huaral to Lima. The transportation took around 2 hours. Once the fruits arrived, they were transferred to the laboratory at Universidad Nacional Agraria La Molina (UNALM). When opening the carboard boxes, it was confirmed the absence of water condensation. Subsequently, the fruits were checked for any defects and they were found to be entire, with firm texture, fresh, with no thorns, cracks, and pest. Fruits were washed, disinfected, rinsed with abundant water, dried, and stored in refrigeration (4 °C) until the next day for processing. In order to detail the transportation, duration after harvesting, and storage conditions in our study, this information was summarized and included in lines 127 – 139. The authors thank the Corporation Abregú in the section of acknowledgments in updated line 769.

- Line120: What’s its final moisture content after drying? Please include. Moreover, how’s about the particle size experiment? At now, it seem likes all peels were milled with the same conditions????

Response: Firstly, we obtained a moisture content for Hylocereus undatus and Hylocereus hybridum peels of 88.2 % and 91.1 % (Table S1 from Supplementary materials), respectively. Nevertheless, we tried to obtain a homogenised result for the final moisture content for both red dragon fruit peel species in order to have comparable results. The final moisture content of Hylocereus undatus and Hylocereus hybridum peels was ⁓5.2 %. Regarding the particle size experiment, we obtained three levels of particle size (coarse, 297 – 499 µm; intermediate, 177 – 296 µm; and fine, <177 µm) through a 3k full factorial design (Figure 1, line 310). After drying, red dragon fruit peels were milled in a rotor beater mill at 3500 rpm, as shown in line 143 – 144. The information regarding the final moisture content of red dragon fruit peels was added in line 142 – 143. Likewise, more details about the fractionation of particle size were added in lines 144 – 147.

- Interestingly, how come of these particle size of the peels? Why author designed by using these 3 sizes? Also, when the author experimented with particle size, what type of red dragon fruit was utilized? (Considering there are two types of this fruit.) If the author used only one type for the particle size experiment, how can they ensure that the outcomes would be similar for the other type?

Response: Thank you for your observation. The motivation for us to investigate the effect of particle size on the colourimetric, physicochemical, and technofunctional properties of red dragon fruit peel powders based on the published article titled “Characteristics of fibre-rich powder and antioxidant activity of pitaya (Hylocereus undatus) peels” (https://doi.org/10.1111/j.1365-2621.2012.02971.x) from Yongliang Zhuang, Yufeng Zhang, and Liping Sun. This study reported the effect of particle size on dietary fiber content, water-holding capacity (WHC), swelling capacity (SC), oil-holding capacity (OHC), among other dependent variables, using three particle size ranges: FRP80 (104 – 178 µm), FRP140 (58 – 104 µm), and FRP250 (<58 µm). The authors showed that the excessive reduction of particle size (i. e., FRP250) significantly affected the soluble, insoluble, total dietary fibre contents, and functional properties (WHC, SC, and OHC). Therefore, we considered the particle size as an important factor for our study, considering greater particle sizes (coarse, 297 – 499 µm; intermediate, 177 – 296 µm; and fine, <177 µm), as indicated in line 144 – 147, and Figure 1 (line 310), in order to avoid negative results during the analyses and further application in the meat product.

Regarding the preliminary experiments evaluating the effect of particle size on the physicochemical and technofunctional properties, we used both Hylocereus undatus and Hylocereus hybridum species. Subsequently, we observed significant differences in the technofunctional (WHC, SC, and OHC) and pectin yield for each of the dragon fruit species.

- Line 124, 225: Change the subtopic into “Color measurement”

Response: Done. The subsections were changed by the phrase “Colour Measurement” in updated lines 149 and 256. Moreover, the phrase was also utilized in the subsections “Colour Measurement of RDF-PP” (line 356) and “Colour Measurement of Emulsified Alpaca-Based Sausages” (line 582).

- Line133: Change the subtopic into "Crude fiber content”

Response: Changed. Now that subsection was modified to “Crude Fibre Content”, shown in line 158.

- Line206: The description of sausage production only covered the process without mentioning the substitution using the peels. Please include details about the experimental sets and the procedure for substituting the peels in the product.

Response: Thank you for your observation. The substitution of pork-back fat was mentioned in lines 244 – 245. However, we have included more details about the experimental sets and fat substitution by red dragon fruit peel powder (RDF-PP) in lines 245 – 250.

- The four formulations of sausages are still confused.

Response: The information about the four formulations of sausages was updated by adding the following text: “Subsequently, pork-back fat substitution with hydrated RDF-PP was performed at 3.29 (F1), 6.57 (F2), and 9.86% (F3) based on the whole formulation (i. e., 100%). Specifically, in order to develop the different fibre-enriched alpaca-based sausage formulations (F1, F2, and F3), in the pre-emulsified mixture, the pork-back fat (15.0%) was replaced by using 21.2 (F1), 42.5 (F2), and 63.7% (F3) RDF-PP, and 0.67 (F1), 1.33 (F2), and 2.00% (F3) water, respectively. The incorporation of RDF-PP and water to the formulations F1, F2, and F3 was based on the WHC of RDF-PP from the best treatment.”, which is in lines 236 – 241.

-Line250: The preparation of sausage before allowing to assess for the scores should be provided. For example, re-heat? What’s size for serving? How’s many samples? What’s time duration during each sample? etc.

Response: Thank you very much for your observation. The requested information about the serving size, sample condition and quantity, evaluation time of samples, among other details, has been included in lines 283 – 287: ‘… for each formulation (C, F1, F2, and F3). Two sausage slices, constituting a serving size of ⁓8 g per formulation, were arranged on a disposable plate and served at room temperature to each panellist. Additionally, each panellist was given a testing time of 9 min and instructed to rinse their mouth with water after evaluating each sample.’

- The statistical analysis section appears excessively long. I recommended that the author condense it by focusing on the main method used.

Response: Thank you for the scope. The introductory part from line 294 – 300 has been removed. However, we consider the information given in the section 2.7.1. ‘Three-level Full Factorial Design and Response Surface Methodology’ (lines 302 – 337) is pertinent to the hypothesis: Do “red dragon fruit peels ratio” and “particle size” influence the colourimetric, physicochemical, and techno-functional properties of red dragon fruit peel powder? which also includes the further optimization of the most important dependent variables (water holding capacity, oil holding capacity, and pectin yield). Likewise, the section 2.7.2. ‘Completely Randomized Design’ (lines 339 – 348) is pertinent to the effect of the best treatment of red dragon fruit peel powder (RDF-PP) on the colourimetric, physicochemical, and texture properties of alpaca-based sausages, and section 2.7.3. ‘Sensory analysis’ (lines 350 – 353) to the comparative study of the reduced-fat alpaca-based sausage formulated with RDF-PP against a control formulation.

RESULTS AND DISCUSSION

- Which of the two species of red dragon fruit peels exhibits better properties? Furthermore, how does the particle size influence these properties? The author should delve into these details as I believe that utilizing an experimental design like this might not allow for the independent separation of each factor's effect.

Response: We appreciate your question. From the two species of red dragon fruit peels, the Hylocereus undatus species exhibited better properties based on the dependent variables water holding capacity (WHC), oil holding capacity (OHC), and pectin yield (PY). The 3k full factorial design allows the modelling and analyses of both the main and interaction effects between the factors: red dragon fruit peels ratio (RDF-PR) and particle size (PS), as shown in Figure S2 (from the Supplementary materials), where the main effects of the optimized independent variables (RDF-PR and PS) are exhibited WHC, OHC, and PY. Likewise, Table 2 (line 425) shows the interaction between RDF-PR and PS on the dependent variables of the study. Regarding the effect of particle size on WHC, additional detail was included in lines 406 – 410. For OHC, the information was reviewed and additional details has been included in lines 479 – 484. For PY, the information pertinent to the effect of particle size is shown in lines 498 – 502.

- The author solely presented the results and compared them with other studies but failed to discuss the underlying reasons affected by the experimental factors. For instance, if 'T1' exhibits the best techno-functional properties, what is the reason behind it? Does it depend on the type of dragon peels or their size? Moreover, there seems to be a lack of specific properties data for individual dragon fruit peel species. Consequently, the results do not conclusively identify their precise origin or characteristics.

Response: Thank you very much for your observation. The results for the most important dependent variables: water holding capacity (WHC), oil holding capacity (OHC), and pectin yield (PY) were discussed in lines 399 – 417, 474 – 484, and 498 – 504, respectively, including additional information as indicated in previous observation. Regarding the reason why ‘T1’ was chosen as the best treatment, a pertinent explanation for WHC, OHC, and PY was given in section 3.1.3. ‘Statistical Validation of Optimal Treatment of RDF-PP’ (lines 542 – 549). Based on this information, it was indicated that T1 meets with the technofunctional and physicochemical characteristics showing ‘the highest WHC, OHC, and PY at a RDF-PR of 100% HU and PS <177 µm (Table 3)’ (lines 548 – 549). Subsequently, considering that ‘T1’ as the optimal treatment, a statistical validation (Table 4) was performed, showing that there was no statistical difference between the observed (experimental) and predicted values (lines 551 – 552).

The response surface methodology allows the evaluation of the main and interaction effects of the factors: red dragon fruit peels ratio (RDF-PR) and particle size (PS). Table 2 (line 425) showed the significant interaction effect between RDF-PR and PS on WHC and PY, as well as the significant main effect of RDF-PR and PS on OHC. On the other hand, the statistical validation permits to corroborate the authenticity of the results of the selected dependent variables based on the levels -1, 0, and 1. For ‘T1’, the work conditions for RDF-PR and PS were at the levels -1 and -1, respectively. Thus, indicating that ‘T1’ depends from RDF-PR and PS.

Regarding the specific properties data for each dragon fruit species, Table S1 (Supplementary materials) shows the characterization data for fruits from Hylocereus undatus and Hylocereus hybridum species. Furthermore, part of the information from Table S1 is indicated in section ‘2.1. Materials’ (from line 123 – 126).

CONCLUSION

- Good.

Response: Thank you for your assessment of our manuscript.

Round 2

Reviewer 2 Report

Comments and Suggestions for Authors

-